# The leap to ordinal: Detailed functional prognosis after traumatic brain injury with a flexible modelling approach

Shubhayu Bhattacharyay[1,2,3]*, Ioan Milosevic[1], Lindsay Wilson[4], David K. Menon[1], Robert D. Stevens[3,5], Ewout W. Steyerberg[6], David W. Nelson[7], Ari Ercole[1,8], the CENTER-TBI investigators participants[¶]

1 Division of Anaesthesia, University of Cambridge, Cambridge, United Kingdom, 2 Department of Clinical Neurosciences, University of Cambridge, Cambridge, United Kingdom, 3 Laboratory of Computational Intensive Care Medicine, Johns Hopkins University, Baltimore, MD, United States of America, 4 Division of Psychology, University of Stirling, Stirling, United Kingdom, 5 Department of Anesthesiology and Critical Care Medicine, Johns Hopkins University, Baltimore, MD, United States of America, 6 Department of Biomedical Data Sciences, Leiden University Medical Center, Leiden, The Netherlands, 7 Department of Physiology and Pharmacology, Section for Perioperative Medicine and Intensive Care, Karolinska Institutet, Stockholm, Sweden, 8 Cambridge Centre for Artificial Intelligence in Medicine, Cambridge, United Kingdom

¶ A full list of the CENTER-TBI investigators and participants can be found in the Acknowledgments.
* sb2406@cam.ac.uk

**Data Availability Statement:** All code used in this project can be found at the following online repository: https://github.com/sbhattacharyay/ordinal_GOSE_prediction (doi: 10.5281/zenodo.

## Abstract

When a patient is admitted to the intensive care unit (ICU) after a traumatic brain injury (TBI), an early prognosis is essential for baseline risk adjustment and shared decision making. TBI outcomes are commonly categorised by the Glasgow Outcome Scale–Extended (GOSE) into eight, ordered levels of functional recovery at 6 months after injury. Existing ICU prognostic models predict binary outcomes at a certain threshold of GOSE (e.g., prediction of survival [GOSE > 1]). We aimed to develop ordinal prediction models that concurrently predict probabilities of each GOSE score. From a prospective cohort ($n$ = 1,550, 65 centres) in the ICU stratum of the Collaborative European NeuroTrauma Effectiveness Research in TBI (CENTER-TBI) patient dataset, we extracted all clinical information within 24 hours of ICU admission (1,151 predictors) and 6-month GOSE scores. We analysed the effect of two design elements on ordinal model performance: (1) the baseline predictor set, ranging from a concise set of ten validated predictors to a token-embedded representation of all possible predictors, and (2) the modelling strategy, from ordinal logistic regression to multinomial deep learning. With repeated $k$-fold cross-validation, we found that expanding the baseline predictor set significantly improved ordinal prediction performance while increasing analytical complexity did not. Half of these gains could be achieved with the addition of eight high-impact predictors to the concise set. At best, ordinal models achieved 0.76 (95% CI: 0.74–0.77) ordinal discrimination ability (ordinal $c$-index) and 57% (95% CI: 54%– 60%) explanation of ordinal variation in 6-month GOSE (Somers' $D_{xy}$). Model performance and the effect of expanding the predictor set decreased at higher GOSE thresholds, indicating the difficulty of predicting better functional outcomes shortly after ICU admission. Our results motivate the search for informative predictors that improve confidence in prognosis of higher GOSE and the development of ordinal dynamic prediction models.

5933042). The minimal data required to reproduce the study's methods, reported statistics, figures, and results can be found among the commented and structured code of this repository. Individual participant data, including data dictionary, the study protocol, and analysis scripts are available online, conditional to approved study proposal, with no end date. Interested investigators must provide a methodologically sound study proposal to the management committee. Proposals can be submitted online at https://www.center-tbi.eu/data. Signed confirmation of a data access agreement is required, and all access must comply with regulatory restrictions imposed on the original study.

**Funding:** The research was supported by the National Institute for Health Research (NIHR) Brain Injury MedTech Co-operative based at Cambridge University Hospitals NHS Foundation Trust and University of Cambridge. The views expressed are those of the author(s) and not necessarily those of the NHS, NIHR or the Department of Health and Social Care. CENTER-TBI was supported by the European Union 7th Framework programme (EC grant 602150). Additional funding was obtained from the Hannelore Kohl Stiftung (Germany), from OneMind (USA), and from Integra LifeSciences Corporation (USA). CENTER-TBI also acknowledges interactions and support from the International Initiative for TBI Research (InTBIR) investigators. CSD3 is supported by the United Kingdom Engineering and Physical Sciences Research Council (EPSRC Tier-2 capital grant EP/T022159/1). SB is currently funded by a Gates Cambridge fellowship. There was no additional external funding received for this study. The funders had no role in study design, data collection and analysis, decision to publish, or preparation of the manuscript.

**Competing interests:** The authors have declared that no competing interests exist.

## Introduction

Globally, traumatic brain injury (TBI) is a major cause of death, disability, and economic burden [1]. The treatment of critically ill TBI patients is largely guided by an initial prognosis made within a day of admission to the intensive care unit (ICU) [2]. Early outcome prediction models set a baseline against which clinicians consider the effect of therapeutic strategies and compare patient trajectories. Therefore, well-calibrated and reliable prognostic models are an essential component of intensive care.

Outcome after TBI is most often evaluated on the ordered, eight-point Glasgow Outcome Scale–Extended (GOSE) [3–6], which stratifies patients by their highest level of functional recovery according to participation in daily activities. Existing baseline prediction models used in the ICU dichotomise the GOSE into binary endpoints for TBI outcome. For example, the Acute Physiologic Assessment and Chronic Health Evaluation (APACHE) II [7] model predicts in-hospital survival (GOSE > 1) while the International Mission for Prognosis and Analysis of Clinical Trials in TBI (IMPACT) [8] models focus on predicting functional independence (GOSE > 4, or 'favourable outcome') and survival at 6 months post-injury.

Dichotomised GOSE prediction employs a fixed threshold of favourability among the eight levels of recovery for all patients. However, there is no empirical justification for an ideal treatment-effect threshold of GOSE [9]. Moreover, dichotomisation removes each patient or caregiver's ability to define a different level of recovery as 'favourable' during prognosis. By concealing the nuanced differences in outcome defined by the GOSE, dichotomisation also limits the prognostic information made available during a shared treatment decision making process. For example, when clinicians, patients, or next of kin must together decide whether to withdraw life-sustaining measures (WLSM) after severe TBI, knowing the probability of different levels of functional recovery in addition to the baseline probability of survival would enable better quality-of-life consideration and confidence in the decision (**Fig 1B**) [10]. These problems of dichotomisation cannot be addressed simply by independently training a combination of binary prediction models at several GOSE thresholds. If model predictions are not constrained across the thresholds (i.e., ensuring probabilities do not increase with higher thresholds) during training, then combining multiple threshold outputs may result in nonsensical values. For example, the purported probability of survival (GOSE > 1) might be lower than that of recovering functional independence (GOSE > 4).

A practical solution would be to train ordinal outcome prediction models, which concurrently return probabilities at each GOSE threshold by learning the interdependent relationships between the predictor set and the possible levels of functional recovery (**Fig 1A**). Ordinal GOSE prediction models would allow users to interpret the probability of different levels of functional recovery. Additionally, they can provide insight into the conditional probability of obtaining greater levels of recovery given lower levels (see **Fig 1B** for a practical clinical application of this information). However, moving from binary to ordinal outcome prediction poses three key challenges. First, there is no guarantee that widely accepted TBI outcome predictor sets, validated either by binary or ordinal regression analysis, will be able to capture the nuanced differences between levels of functional recovery well enough for reliable prediction. Second, ordinal prediction models typically need to be more complicated than binary models to encode the possibility of more outcomes and the constrained relationship between them [11]. For GOSE prediction, ordinal models can either encode the outcomes as: (1) multinomial, in which nodes exist for each GOSE score and collectively undergo a softmax transformation (to constrain the sum of values to one) and probabilities are calculated by accumulating values up to each threshold, or (2) ordinal, in which nodes exist for each threshold between consecutive GOSE scores, constrained such that output values must not increase

## A

### Binary prediction models

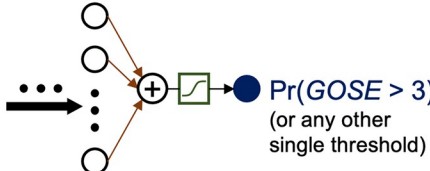

### Ordinal prediction models

*Multinomial outcome encoding*

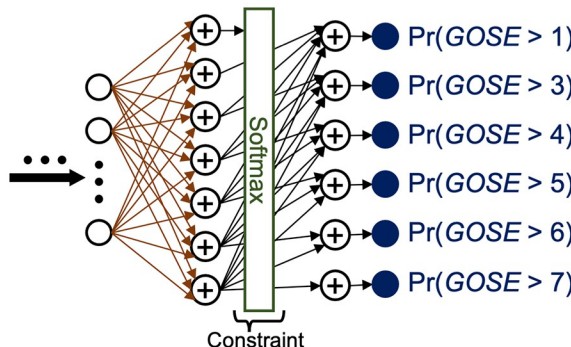

*Ordinal outcome encoding*

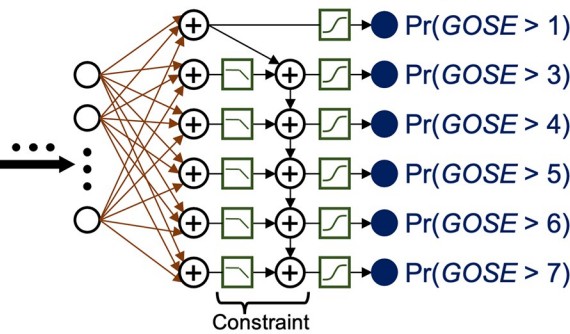

### Legend

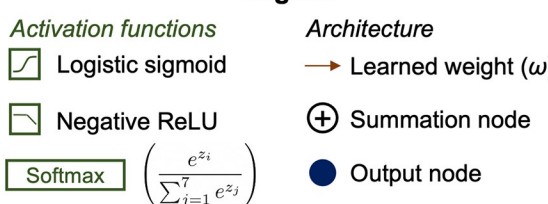

## B

### Sample patient case

***Presentation***:

- Severe traumatic brain injury
- On life-sustaining therapy in the ICU
- Family would strongly prefer to withdraw from life-sustaining therapy if patient is not expected to regain conscious, partial functional independence (*GOSE* > 3) within 6 months

### Baseline prognosis with binary prediction model

***Model output***:

Pr(*GOSE* > 3) = 0.1228617

***Interpretation***:

"The patient has an 12.3% chance of recovering conscious, partial functional independence within 6 months."

### Baseline prognosis with ordinal prediction model

***Model output***:

| | |
|---|---|
| Pr(*GOSE* > 1) | = 0.1273615 |
| Pr(*GOSE* > 3) | = 0.1228617 |
| Pr(*GOSE* > 4) | = 0.0661974 |
| Pr(*GOSE* > 5) | = 0.0261596 |
| Pr(*GOSE* > 6) | = 0.0216245 |
| Pr(*GOSE* > 7) | = 0.0038411 |

***Interpretation 1***:

"The patient has a 12.7% chance of survival up to 6 months and a 12.3% chance of recovering conscious, partial functional independence within 6 months."

***Bespoke conditional probability diagram***:

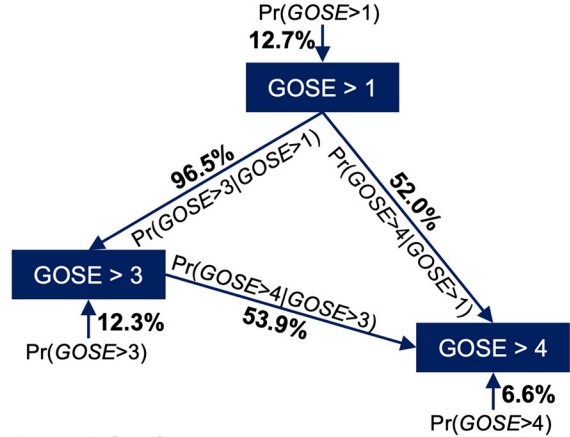

***Interpretation 2***:

"If the patient does survive up to 6 months, they have a 96.5% chance of recovering conscious, partial functional independence and a 52.0% chance of regaining full functional independence."

**Fig 1. Comparison of ordinal outcome prediction to binary outcome prediction in terms of model architecture and clinical application.** GOSE = Glasgow Outcome Scale–Extended at 6 months post-injury. ReLU = rectified linear unit. Pr(•) = Probability operator, i.e., "probability of •." Pr(•|○) = Conditional probability operator, i.e., "probability of •, given ○." (**A**) Output layer architectures of binary and ordinal GOSE prediction models. Ordinal prediction models

must not only have a more complicated output structure (in terms of learned weights and outcome encoding choices) but also constrain probabilities across the possible levels of functional outcome (indicated by 'Constraint' in the ordinal model representations). The constraint for multinomial outcome encoding is performed with a softmax activation function while the constraint for ordinal outcome encoding is performed with subtractions of output values (implemented with a negative ReLU transformation) from lower thresholds. In the provided legend formula for the softmax activation function, $z_i$ represents the outputted value of the $i^{th}$ node of the multinomial outcome encoding layer (i.e., the node representing the $i^{th}$ possible score of GOSE) preceding the softmax transformation. (**B**) A sample patient case to demonstrate the difference in prognostic information between ordinal and binary GOSE prediction models. Binary models predict outcomes at one GOSE threshold while ordinal models predict outcomes at every GOSE threshold concurrently and provide conditional predictions of higher GOSE threshold outcomes given lower GOSE threshold outcomes. Bespoke conditional probability diagrams can be constructed between any number of GOSE thresholds, as desired by model users, so long as lower thresholds (e.g., GOSE > 1) precede higher thresholds (e.g., GOSE > 3) in directionality. Conditional probabilities are calculated by dividing the model probability at the higher threshold by the model probability at the lower threshold (e.g., $\Pr(GOSE > 3 | GOSE > 1) = \Pr(GOSE > 3)/\Pr(GOSE > 1)$).

with higher thresholds, and probabilities for each threshold are calculated with a sigmoid transformation (**Fig 1A**). Third, assessment of prediction performance is not as intuitive with an ordinal outcome as with a binary outcome. Widely used dichotomous prediction performance metrics such as the *c*-index (i.e., the area under the receiver operating characteristic curve [AUC]) do not trivially extend to the ordinal case [12], so assessment of ordinal prediction models requires the consideration of multifactorial metrics and visualisations that may complicate interpretations of model performance [13].

As part of the Collaborative European NeuroTrauma Effectiveness Research in TBI (CEN-TER-TBI) project, we aim to address the challenges of ordinal outcome prediction. Our analyses cover a range of modelling strategies and predictors available within the first 24 hours of admission to the ICU.

## Materials and methods

### Study population and dataset

The study population was extracted from the ICU stratum of the core CENTER-TBI dataset (v3.0) using Opal database software [14]. The project objectives and experimental design of CENTER-TBI have been described in detail by Maas *et al.* [15] and Steyerberg *et al.* [16] Study patients were prospectively recruited at one of 65 participating ICUs across Europe with the following eligibility criteria: admission to the hospital within 24 hours of injury, indication for CT scanning, and informed consent according to local and national requirements.

Per project protocol, each patient's follow-up schedule included a GOSE assessment at 6 months post-injury, or, more precisely, within a window of 5–8 months post-injury. GOSE assessments were conducted using structured interviews [6] and patient/carer questionnaires [17] by the clinical research team of CENTER-TBI. The eight, ordinal scores of GOSE, representing the highest levels of functional recovery, are decoded in the heading of **Table 1**. Since patient/carer questionnaires do not distinguish vegetative patients (GOSE = 2) into a separate category, GOSE scores 2 and 3 (lower severe disability) were combined to one category (GOSE $\in \{2,3\}$) in our dataset. Of the 2,138 ICU patients in the CENTER-TBI dataset available for analysis, we excluded patients in the following order: (1) age less than 16 years at ICU admission ($n = 82$), (2) follow-up GOSE was unavailable ($n = 283$), and (3) ICU stay was less than 24 hours ($n = 223$). Our resulting sample size was $n = 1,550$. For 1,351 patients (87.2%), either the patient died during ICU stay ($n = 205$) or results from a GOSE evaluation at 5–8 months post-injury were available in the dataset ($n = 1,146$). For the remaining 199 patients (12.8%), GOSE scores were imputed using a Markov multi-state model based on the observed GOSE scores recorded at different timepoints between 2 weeks to one-year post-injury [18]. A flow diagram for study inclusion and follow-up is provided in **S1 Fig**, and summary characteristics of the study population are detailed in **Table 1**.

**Table 1. Summary characteristics of the study population at ICU admission stratified by ordinal 6-month outcomes.**

| Summary characteristics | Overall | Glasgow Outcome Scale–Extended (GOSE) at 6 months post-injury | | | | | | | p-value[‡] |
|---|---|---|---|---|---|---|---|---|---|
| | | (1) Death | (2 or 3) Vegetative or lower severe disability | (4) Upper severe disability | (5) Lower moderate disability | (6) Upper moderate disability | (7) Lower good recovery | (8) Upper good recovery | |
| n[*] | 1550 | 318 (20.5%) | 262 (16.9%) | 120 (7.7%) | 227 (14.6%) | 200 (12.9%) | 206 (13.3%) | 217 (14.0%) | |
| Age [years] | 51 (31–66) | 66 (50–76) | 55 (36–68) | 48 (29–61) | 44 (31–56) | 41 (27–53) | 48 (31–65) | 41 (24–61) | <0.0001 |
| Sex | | | | | | | | | 0.59 |
| Female | 409 (26.4%) | 78 (24.5%) | 71 (27.1%) | 43 (35.8%) | 64 (28.2%) | 49 (24.5%) | 59 (28.6%) | 45 (20.7%) | |
| Race (n[†] = 1427) | | | | | | | | | 0.13 |
| White | 1386 (97.1%) | 281 (97.2%) | 239 (96.8%) | 106 (95.5%) | 195 (96.5%) | 183 (97.3%) | 184 (98.4%) | 198 (97.5%) | |
| Black | 21 (1.5%) | 2 (0.7%) | 4 (1.6%) | 3 (2.7%) | 5 (2.5%) | 3 (1.6%) | 2 (1.1%) | 2 (1.0%) | |
| Asian | 20 (1.4%) | 6 (2.1%) | 4 (1.6%) | 2 (1.8%) | 2 (1.0%) | 2 (1.1%) | 1 (0.5%) | 3 (1.5%) | |
| Baseline GCS (n[†] = 1465) | 8 (4–14) | 5 (3–10) | 6 (3–10) | 8 (4–13) | 8 (5–13) | 9 (6–14) | 13 (7–15) | 13 (8–15) | <0.0001 |
| Mild [13–15] | 390 (26.6%) | 30 (10.3%) | 38 (15.3%) | 26 (23.4%) | 42 (19.5%) | 66 (34.9%) | 91 (45.3%) | 97 (46.4%) | |
| Moderate [9–12] | 331 (22.6%) | 65 (22.3%) | 41 (16.5%) | 28 (25.2%) | 65 (30.2%) | 36 (19.0%) | 40 (19.9%) | 56 (26.8%) | |
| Severe [3–8] | 744 (50.8%) | 196 (67.4%) | 170 (68.3%) | 57 (51.4%) | 108 (50.2%) | 87 (46.0%) | 70 (34.8%) | 56 (26.8%) | |

Data are median (IQR) for continuous characteristics and n (% of column group) for categorical characteristics, unless otherwise indicated. Units or numerical definitions of characteristics are provided in square brackets. Baseline GCS = Glasgow Coma Scale at ICU admission, from 3 to 15. Conventionally, TBI severity is categorically defined by baseline GCS scores as indicated in square brackets.

[*]Percentages for sample size (n) represent proportion of study sample size in each GOSE group.

[†]Limited sample size of non-missing values for characteristic.

[‡]p-values are determined from proportional odds logistic regression (POLR) coefficient analysis trained on all summary characteristics concurrently [19]. For categorical variables with $k > 2$ categories (e.g., Race), p-values were calculated with a likelihood ratio test (with $k$-1 degrees of freedom) on POLR.

## Repeated *k*-fold cross-validation

We implemented the 'scikit-learn' module (v0.23.2) [20] in Python (v3.7.6) to create 100 stratified partitions of our study population for repeated *k*-fold cross-validation (20 repeats, 5 folds). Within each of the partitions, approximately 80% of the population would constitute the training set ($n \approx 1{,}240$ patients) and 20% of the population would constitute the corresponding testing set ($n \approx 310$ patients). For parametric (i.e., deep learning) models, we implemented a stratified shuffle split on each of the 100 training sets to set 15% ($n \approx 46$ patients) aside for validation and hyperparameter optimisation.

## Selection and preparation of concise predictor set

In selecting a concise predictor set, our primary aim was to find a small group of well-validated, widely measured clinical variables that are commonly used for TBI outcome prognosis in existing ICU practice. We selected the ten predictors from the extended IMPACT binary prediction model [8] for moderate-to-severe TBI–defined by a baseline Glasgow Coma Scale (GCS) [21, 22] score between 3 and 12, inclusive–to represent our concise set. While 26.6% of our study population falls out of this GCS range (**Table 1**), we find that the IMPACT predictor

**Table 2. Concise baseline predictors of the study population stratified by ordinal 6-month outcomes.**

| Concise predictors | Overall (n = 1550) | Glasgow Outcome Scale–Extended (GOSE) at 6 months post-injury | | | | | | | p-value[‡] |
| --- | --- | --- | --- | --- | --- | --- | --- | --- | --- |
| | | 1 (n = 318) | 2 or 3 (n = 262) | 4 (n = 120) | 5 (n = 227) | 6 (n = 200) | 7 (n = 206) | 8 (n = 217) | |
| Age [years] | 51 (31–66) | 66 (50–76) | 55 (36–68) | 48 (29–61) | 44 (31–56) | 41 (27–53) | 48 (31–65) | 41 (24–61) | <0.0001 |
| GCSm ($n^†$ = 1509) | 5 (1–6) | 2 (1–5) | 3 (1–5) | 5 (1–6) | 5 (1–6) | 5 (2–6) | 5 (3–6) | 6 (5–6) | <0.0001 |
| (1) No response | 484 (32.1%) | 152 (50.0%) | 104 (40.6%) | 35 (29.9%) | 63 (28.5%) | 46 (23.6%) | 47 (23.0%) | 37 (17.5%) | |
| (2) Abnormal extension | 54 (3.6%) | 17 (5.6%) | 20 (7.8%) | 4 (3.4%) | 6 (2.7%) | 3 (1.5%) | 2 (1.0%) | 2 (0.9%) | |
| (3) Abnormal flexion | 63 (4.2%) | 14 (4.6%) | 12 (4.7%) | 8 (6.8%) | 11 (5.0%) | 8 (4.1%) | 4 (2.0%) | 6 (2.8%) | |
| (4) Withdrawal from stimulus | 114 (7.6%) | 27 (8.9%) | 23 (9.0%) | 8 (6.8%) | 20 (9.0%) | 21 (10.8%) | 8 (3.9%) | 7 (3.3%) | |
| (5) Movement localised to stimulus | 305 (20.2%) | 52 (17.1%) | 47 (18.4%) | 24 (20.5%) | 50 (22.6%) | 46 (23.6%) | 44 (21.6%) | 42 (19.8%) | |
| (6) Obeys commands | 489 (32.4%) | 42 (13.8%) | 50 (19.5%) | 38 (32.5%) | 71 (32.1%) | 71 (36.4%) | 99 (48.5%) | 118 (55.7%) | |
| Unreactive pupils ($n^†$ = 1465) | | | | | | | | | <0.0001 |
| One | 111 (7.6%) | 31 (10.5%) | 31 (12.3%) | 7 (6.3%) | 20 (9.3%) | 5 (2.6%) | 8 (4.1%) | 9 (4.4%) | |
| Two | 168 (11.5%) | 84 (28.5%) | 33 (13.0%) | 8 (7.2%) | 14 (6.5%) | 8 (4.2%) | 16 (8.2%) | 5 (2.4%) | |
| Hypoxia | 207 (13.4%) | 60 (18.9%) | 33 (12.6%) | 14 (11.7%) | 35 (15.4%) | 33 (16.5%) | 16 (7.8%) | 16 (7.4%) | 0.37 |
| Hypotension | 210 (13.5%) | 56 (17.6%) | 51 (19.5%) | 21 (17.5%) | 32 (14.1%) | 22 (11.0%) | 15 (7.3%) | 13 (6.0%) | 0.0038 |
| Marshall CT ($n^†$ = 1255) | VI (II–VI) | III (II–VI) | II (II–VI) | II (II–VI) | II (II–II) | II (II–III) | II (II–II) | VI (II–VI) | 0.043 |
| No visible pathology (I) | 118 (9.4%) | 8 (3.3%) | 11 (5.3%) | 5 (5.2%) | 17 (8.7%) | 25 (15.2%) | 24 (13.6%) | 28 (16.5%) | |
| Diffuse injury II | 592 (47.2%) | 56 (22.8%) | 84 (40.6%) | 54 (56.2%) | 92 (47.2%) | 100 (60.6%) | 103 (58.5%) | 103 (60.6%) | |
| Diffuse injury III | 108 (8.6%) | 42 (17.1%) | 17 (8.2%) | 10 (10.4%) | 14 (7.2%) | 9 (5.5%) | 6 (3.4%) | 10 (5.9%) | |
| Diffuse injury IV | 16 (1.3%) | 7 (2.8%) | 1 (0.5%) | 1 (1.0%) | 4 (2.1%) | 1 (0.6%) | 1 (0.6%) | 1 (0.6%) | |
| Mass lesion (V & VI) | 421 (33.5%) | 133 (54.0%) | 94 (45.4%) | 26 (27.1%) | 68 (34.9%) | 30 (18.2%) | 42 (23.9%) | 28 (16.5%) | |
| tSAH ($n^†$ = 1254) | 957 (76.3%) | 221 (90.2%) | 176 (84.2%) | 73 (76.0%) | 150 (76.9%) | 106 (63.9%) | 125 (71.4%) | 106 (63.1%) | 0.16 |
| EDH ($n^†$ = 1257) | 244 (19.4%) | 31 (12.7%) | 32 (15.3%) | 21 (21.9%) | 46 (23.6%) | 32 (19.3%) | 42 (23.9%) | 40 (23.5%) | 0.016 |
| Glucose [mmol/L] ($n^†$ = 1062) | 7.7 (6.6–9.4) | 8.8 (7.3–11) | 8.0 (6.5–9.8) | 7.6 (6.5–9.3) | 7.8 (6.6–9.6) | 7.7 (6.5–8.7) | 7.3 (6.3–8.5) | 7.1 (6.3–8.1) | 0.013 |
| Hb [g/dL] ($n^†$ = 1140) | 13 (12–14) | 13 (11–14) | 13 (11–14) | 14 (12–14) | 13 (12–14) | 14 (12–15) | 13 (12–15) | 14 (13–15) | 0.038 |

Data are median (IQR) for continuous characteristics and n (% of column group) for categorical characteristics. Units of characteristics are provided in square brackets. GCSm = motor component score of the Glasgow Coma Scale. Marshall CT = Marshall computerised tomography classification. tSAH = traumatic subarachnoid haemorrhage. EDH = extradural haematoma. Hb = haemoglobin.

[†]Limited sample size of non-missing values for characteristic.

[‡]p-values are determined from proportional odds logistic regression (POLR) analysis trained on all concise predictors concurrently [19] and are combined across 100 missing value imputations via z-transformation [29]. For categorical variables with $k > 2$ categories (e.g., GCSm), p-values were calculated with a likelihood ratio test (with $k$-1 degrees of freedom) on POLR.

set is the most rigorously validated [23–27] baseline set available for the overall critically ill TBI population. The ten predictors, characterised in **Table 2**, are all measured within 24 hours of ICU admission and include demographic characteristics, clinical severity scores, CT characteristics, and laboratory measurements. The predictors as well as empirical justification for their inclusion in the IMPACT model have been described in detail [28]. In this manuscript, each of the models trained on the IMPACT predictor set is denoted as a concise-predictor-based model (CPM).

Seven of the concise predictors had missing values for some of the patients in our study population (**S2 Fig**). In each repeated cross-validation partition, we trained an independent, stochastic predictive mean matching imputation function on the training set and imputed all

missing values across both sets using the 'mice' package (v3.9.0) [30] in R (v4.0.0) [31]. The result was a multiply imputed ($m$ = 100) dataset with a unique imputation per partition, allowing us to simultaneously account for the variability due to resampling and the variability due to missing value imputation during repeated cross-validation.

Prior to the training of CPMs, each of the multi-categorical variables (i.e., GCSm, Marshall CT, and unreactive pupils in **Table 2**) were one-hot encoded and each of the continuous variables (i.e., age, glucose, and haemoglobin) were standardised based on the mean and standard deviation of each of the training sets with the 'scikit-learn' module in Python.

## Selection of concise-predictor-based models (CPMs)

We tested four CPM types, each denoted by a subscript: (1) multinomial logistic regression (CPM$_{MNLR}$), (2) proportional odds (i.e., ordinal) logistic regression (CPM$_{POLR}$), (3) class-weighted feedforward neural network with a multinomial (i.e., softmax) output layer (CPM$_{DeepMN}$), and (4) class-weighted feedforward neural network with an ordinal (i.e., constrained sigmoid at each threshold) output layer (CPM$_{DeepOR}$). These models were selected because, in the setting of ordinal GOSE prediction, we wished to compare the performance of: (1) nonparametric logistic regression models (CPM$_{MNLR}$ and CPM$_{POLR}$) to nonlinear, parametric deep learning networks (CPM$_{DeepMN}$ and CPM$_{DeepOR}$), and (2) multinomial outcome encoding (CPM$_{MNLR}$ and CPM$_{DeepMN}$) to ordinal outcome encoding (CPM$_{POLR}$ and CPM$_{DeepOR}$). Each of these model types returns a predicted probability for each of the GOSE thresholds at 6 months post-injury from the concise set of predictors (**Fig 1A**). A detailed explanation of CPM architectures, hyperparameters for the parametric CPMs, loss functions, and optimisation algorithms is provided in **S1 Appendix**.

CPM$_{Best}$ denotes the optimal CPM for a given performance metric in the **Results**. CPM$_{MNLR}$ and CPM$_{POLR}$ were implemented with the 'statsmodels' module (dev. v0.14.0) [32] in Python, and CPM$_{DeepMN}$ and CPM$_{DeepOR}$ were implemented with the 'PyTorch' (v1.10.0) [33] module in Python.

## Design of all-predictor-based models (APMs)

In contrast to the CPMs, we designed and trained prediction models on all baseline (i.e., available to ICU clinicians at 24 hours post-admission) clinical information (excluding high-resolution data such as full brain images or physiological waveforms) in the CENTER-TBI database. Each of these models is designated as an all-predictor-based model (APM).

For our study population, there are 1,151 predictors [34], each being in one of the 14 categories listed in **Table 3**, with variable levels of missingness and frequency per patient. This information also includes 81 predictors denoting treatments or interventions within the first 24 hours of ICU care (e.g., type and dose of medication administered) and 76 predictors denoting the explicit impressions or rationales of ICU physicians (e.g., reason for surgical intervention and expected prognosis with or without surgery).

To prepare this information into a suitable format for training APMs, we tokenised and embedded heterogenous patient data [35] in a process visualised in **Fig 2**. Predictor tokens were constructed in one of the following ways: (1) for categorical predictors, a token was constructed by concatenating the predictor name and value, e.g., 'GCSTotalScore_04,' (2) for continuous predictors, a token was constructed by learning the distribution of that predictor from the training set and discretising into 20 quantile bins, e.g., 'SystolicBloodPressure_BIN17,' (3) for text-based entries, we removed all special characters, spaces, and capitalisation from the text and appended the unformatted text to the predictor name, e.g., 'InjuryDescription_skullfracture,' and (4) for missing values, a separate token was created to designate missingness,

Table 3. Predictor baseline tokens per patient in the CENTER-TBI dataset.

| Predictor category | Types of tokens | | | | |
|---|---|---|---|---|---|
| | All | Fixed at ICU admission | Continuous variable | Treatments and interventions | Physician impression or rationale |
| Emergency care and ICU admission | 112 (103–121) | 112 (103–121) | 13 (10–16) | 0 (0–0) | 7 (7–8) |
| Brain imaging | 94 (72–114) | 74 (68–83) | 5 (2–8) | 0 (0–0) | 9 (8–10) |
| ICU monitoring and management | 63 (52–72) | 3 (3–3) | 10 (5–13) | 40 (34–46) | 13 (3–15) |
| Injury characteristics and severity | 55 (49–62) | 55 (49–62) | 2 (2–2) | 0 (0–0) | 0 (0–0) |
| End-of-day assessments | 50 (45–54) | 0 (0–0) | 19 (17–21) | 0 (0–0) | 0 (0–0) |
| Laboratory measurements | 44 (32–55) | 14 (0–20) | 42 (31–52) | 0 (0–0) | 1 (1–1) |
| Medical and behavioural history | 38 (32–51) | 38 (32–51) | 0 (0–1) | 0 (0–0) | 0 (0–0) |
| Medications | 30 (21–40) | 0 (0–0) | 0 (0–0) | 22 (15–30) | 8 (5–11) |
| Bihourly assessments | 17 (0–32) | 0 (0–0) | 15 (0–27) | 1 (0–2) | 0 (0–0) |
| Demographics and socioeconomic status | 15 (14–16) | 15 (14–16) | 2 (1–2) | 0 (0–0) | 0 (0–0) |
| Protein biomarkers | 5 (5–5) | 0 (0–0) | 5 (5–5) | 0 (0–0) | 0 (0–0) |
| Surgery | 2 (1–6) | 1 (1–2) | 0 (0–0) | 0 (0–1) | 1 (0–3) |
| Haemostatic markers* | 0 (0–0) | 0 (0–0) | 0 (0–0) | 0 (0–0) | 0 (0–0) |
| Transitions of care* | 0 (0–0) | 0 (0–0) | 0 (0–0) | 0 (0–0) | 0 (0–0) |
| **All predictors** | 532 (486–580) | 315 (288–341) | 111 (90–132) | 64 (50–75) | 37 (29–44) |

Data represent median (IQR) number of non-missing, unique tokens per patient. Tokens were extracted from the clinical information available up to 24 hours after ICU admission for each study patient in the Collaborative European NeuroTrauma Effectiveness Research in TBI (CENTER-TBI) project dataset. Each token may be of only one predictor category (leftmost column) and of any number of token types (four rightmost columns). ICU = intensive care unit.

*Due to their relative infrequency in the CENTER-TBI dataset, these baseline predictor categories have a $3^{rd}$ quartile of zero tokens per patient.

e.g., 'PriorMedications_NA' (**Fig 2A**). The unique tokens from a patient's first 24 hours of ICU stay made up his or her individual predictor set, and the median number of unique tokens (excluding missing value tokens) per patient per predictor category are provided in **Table 3**. Notably, this process does not require any data cleaning, missing value imputation, outlier removal, or domain-specific knowledge for a large set of variables and imposes no constraints on the number or type of predictors per patients [35]. Additionally, by including missing value tokens, models can discover meaningful patterns of missingness if they exist [36].

Taking inspiration from artificially intelligent (AI) natural language processing [37, 38], all the predictor tokens from the training set (excluding the validation set) are used to construct a token dictionary. APMs learn a lower dimensional vector as well as a positive significance weight for each entry in the dictionary during training. The vectors for each of the tokens of a single patient are significance-weight-averaged into a single vector which is then fed into a class-weighted feedforward neural network (**Fig 2B**). If the neural network has no hidden layers, then the APM is analogous to logistic regression, while if it does have hidden layers, the APM corresponds to deep learning. In this work, we train APMs with one of two kinds of output layers: multinomial, i.e., softmax, ($APM_{MN}$), or ordinal, i.e., constrained sigmoid at each GOSE threshold, ($APM_{OR}$). Both model types output a predicted probability for each of the GOSE thresholds at 6 months post-injury. A detailed explanation of APM architectures, hyperparameters, loss functions, and optimisation algorithms is provided in **S2 Appendix**.

$APM_{Best}$ denotes the optimal APM for a given performance metric in the **Results**. $APM_{MN}$ and $APM_{OR}$ were implemented with the 'PyTorch' module in Python.

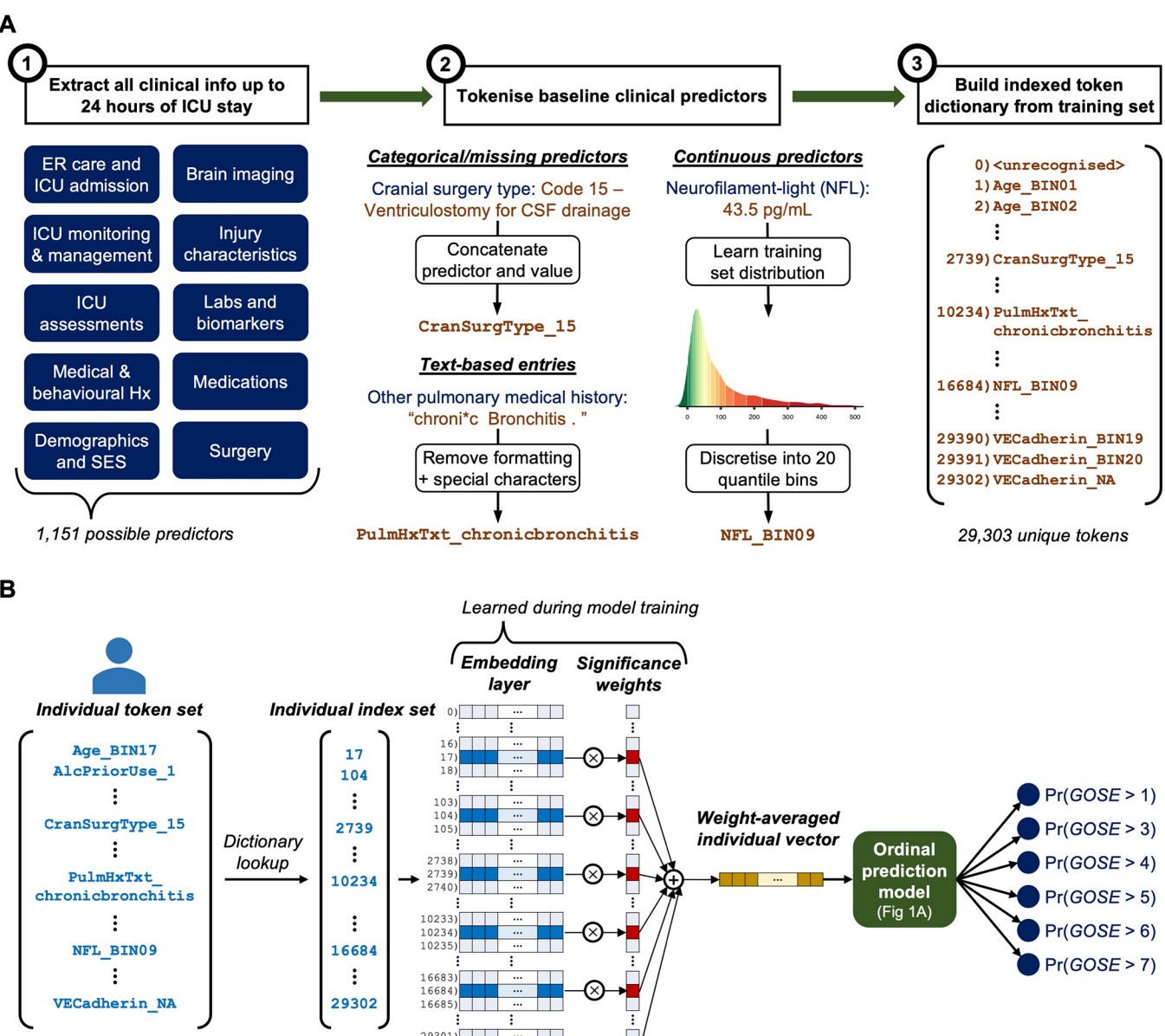

**Fig 2. Tokenisation and embedding procedure for the development of ordinal all-predictor-based models (APMs).** ICU = intensive care unit. ER = emergency room. Hx = history. SES = socioeconomic status. CSF = cerebrospinal fluid. GOSE = Glasgow Outcome Scale–Extended at 6 months post-injury. (**A**) Process of converting all clinical information, from the first 24 hours of each patient, into an indexed dictionary of tokens during model training. The tokenisation process is illustrated with three example predictors and their associated values in step 2. The first entry in the trained token dictionary ('0) <unrecognised>') of step 3 is a placeholder token for any tokens encountered in the testing set that were not seen in the training set. (**B**) Visual representation of token embedding and significance-weighted averaging pipeline during APM prediction runs. After tokenising an individual patient's clinical information, the vector of tokens is converted to a vector of the indices corresponding to each token in the trained token dictionary. The corresponding vectors and significance weights of the indices are extracted to weight-average the patient information into a single vector. The embedding layer and significance weights are learned through stochastic gradient descent during model training, and significance weights are constrained to be positive with an exponential function. While not explicitly shown, the weighted vectors are divided by the number of vectors during weight-averaging. The individual, weight-averaged vector then feeds into an ordinal prediction model to return probabilities at each GOSE threshold. The ordinal prediction model could either have multinomial output encoding (APM_MN) or ordinal outcome encoding (APM_OR), as represented in **Fig 1A**.

## Predictor importance in all-predictor-based models (APMs)

The relative importance of predictor tokens in the trained APMs was measured with absolute Shapley additive explanation (SHAP) [39] values, which, in our case, can be interpreted as the

magnitude of the relative contribution of a token towards a model output for a single patient. For $APM_{MN}$, this corresponds to the predictor contributions towards each node (after softmax transformation, **Fig 1A**) corresponding to the probability at a GOSE score. For $APM_{OR}$, this corresponds to the predictor contributions towards each node (after sigmoid transformation, **Fig 1A**) corresponding to the probability at a GOSE threshold. Absolute SHAP values were measured for each patient in the testing set of every repeated cross-validation partition, and we averaged these values over the partitions to derive our individualised importance scores per token. These scores were averaged, once again, over the entire patient set to calculate the mean absolute SHAP values of each token. Finally, to derive importance scores for each predictor, we calculated the maximum of the mean absolute SHAP values of the possible tokens from the predictor.

## Selection and preparation of extended concise predictor set

We selected a small set of the most important APM predictors by mean absolute SHAP values to add to the concise predictor set and observe the change in model performance. Since the concise predictor set does not include any information on intervention decisions or physician impressions from the first day, we did not consider these predictor types. Moreover, for every multi-categorical predictor selected, we examined the mean absolute SHAP values of each of the predictor's possible tokens to determine which of the categories should be explicitly encoded (e.g., including 10 categories for employment status or just one indicator variable for retirement). The extended concise predictor set, including the 10 original concise predictors and the 8 added predictors, in our study population is listed and characterised in **S1 Table**. Each of the models trained on the concise set with these variables added is denoted as an extended concise-predictor-based model (eCPM).

The process of multiple imputation ($m = 100$), one-hot encoding, and standardisation of the extended concise predictor set was identical to that of the concise predictor set, as described earlier.

## Selection of extended concise-predictor-based models (eCPMs)

The four eCPM model types we tested are identical to the four CPM model types, as described earlier and in **S1 Appendix** with, however, the extended concise predictor set: (1) multinomial logistic regression ($eCPM_{MNLR}$), (2) proportional odds (i.e., ordinal) logistic regression ($eCPM_{POLR}$), (3) class-weighted feedforward neural network with a multinomial (i.e., softmax) output layer ($eCPM_{DeepMN}$), and (4) class-weighted feedforward neural network with an ordinal (i.e., constrained sigmoid at each threshold) output layer ($eCPM_{DeepOR}$).

$eCPM_{Best}$ denotes the optimal eCPM for a given performance metric in the **Results**.

## Assessment of model discrimination and calibration

All model metrics, curves, and associated confidence intervals (CI) were calculated from testing set predictions using the repeated Bootstrap Bias Corrected Cross-Validation (BBC-CV) method [40] with 1,000 resamples of unique patients for bootstrapping. The collection of metrics from the bootstrapped testing set resamples for each model then formed our unbiased estimation distribution for statistical inference (i.e., CI).

In this work, we assess model discrimination performance (i.e., how well do the models separate patients with different GOSE scores?) and probability calibration (i.e., how reliable are the predicted probabilities at each threshold?). The metrics and visualisations are explained in detail, with mathematical derivation and intuitive examples, in **S3 Appendix**. In this section, we will only list the metrics, their interpretations, and their range of feasible values. Feasible

values range from the value corresponding to no model information or random guessing (i.e., the no information value [NIV]) to the value corresponding to ideal model performance (i.e., the full information value [FIV]).

Our primary metric of model discrimination performance is the ordinal $c$-index (ORC) [13]. ORC has two interpretations: (1) the probability that a model correctly separates two patients with two randomly chosen GOSE scores and (2) the average proportional closeness between a model's functional outcome ranking of a set of patients (which includes one randomly chosen patient from each possible GOSE score) to their true functional outcome ranking. In addition, we calculate Somers' $D_{xy}$ [41, 42], which is interpreted as the proportion of ordinal variation in GOSE that can be explained by the variation in model output. Our final metrics of model discrimination are dichotomous $c$-indices (i.e., AUC) at each threshold of GOSE. Each is interpreted as the probability of a model correctly discriminating a patient with GOSE above the threshold from one with GOSE below. The range of feasible values for each discrimination metric are: $\text{NIV}_{\text{ORC}} = 0.5$ to $\text{FIV}_{\text{ORC}} = 1$, $\text{NIV}_{\text{Somers' } Dxy} = 0$ to $\text{FIV}_{\text{Somers' } Dxy} = 1$, and $\text{NIV}_{\text{Dichotomous } c\text{-index}} = 0.5$ to $\text{FIV}_{\text{Dichotomous } c\text{-index}} = 1$. ORC is the only discrimination metric that is independent of the sample prevalence of each GOSE category [13].

To assess the calibration of predicted probabilities at each GOSE threshold, we use the logistic recalibration framework [43] to measure calibration slope [44]. A calibration slope less than one indicates overfitting (i.e., high predicted probabilities are overestimated while low predicted probabilities are underestimated) while a calibration slope greater than one indicates underfitting [45]. We also examine smoothed probability calibration curves [46] to detect miscalibrations that may be overlooked by the logistic recalibration framework [45]. The ideal calibration curve is a diagonal line with slope one and $y$-intercept 0 while one indicative of random guessing would be a horizontal line with a $y$-intercept at the proportion of the study population above the given threshold. We accompany each calibration curve with the integrated calibration index (ICI) [47], which is the mean absolute error between the smoothed and the ideal calibration curves, to aid comparison of curves across model types. $\text{FIV}_{\text{ICI}} = 0$, but $\text{NIV}_{\text{ICI}}$ varies based on the outcome distribution at each threshold (**S3 Appendix**).

All metrics were calculated using the 'scikit-learn' and 'SciPy' (v1.6.2) [48] modules in Python and figures were plotted using the 'ggplot2' package (v3.3.2) [49] in R.

## Computational resources

All computational and statistical components of this work were performed in parallel on the Cambridge Service for Data Driven Discovery (CSD3) high performance computer, operated by the University of Cambridge Research Computing Service (http://www.hpc.cam.ac.uk). The training of each APM was accelerated with graphical processing units and the 'PyTorch Lightning' (v1.5.0) [50] module. The training of all parametric models ($\text{CPM}_{\text{DeepMN}}$, $\text{CPM}_{\text{DeepOR}}$, $\text{APM}_{\text{MN}}$, $\text{APM}_{\text{OR}}$, $\text{eCPM}_{\text{DeepMN}}$, and $\text{eCPM}_{\text{DeepOR}}$) was made more efficient by dropping out consistently underperforming parametric configurations, on the validation sets, with the Bootstrap Bias Corrected with Dropping Cross-Validation (BBCD-CV) method [40] with 1,000 resamples of unique patients. The results of hyperparameter optimisation are detailed in **S4 Appendix**.

## Results

### CPM and APM discrimination performance

The discrimination performance metrics for each CPM are listed in **S2 Table**. Deep learning models ($\text{CPM}_{\text{DeepMN}}$ and $\text{CPM}_{\text{DeepOR}}$) made no significant improvement (based on 95% CI) over logistic regression models ($\text{CPM}_{\text{MNLR}}$ and $\text{CPM}_{\text{POLR}}$). The only significant difference in

**Table 4. Best ordinal model discrimination and calibration performance per predictor set.**

| Metric | Threshold | Model | | |
|---|---|---|---|---|
| | | CPM_Best | APM_Best | eCPM_Best |
| Ordinal $c$-index (ORC) | | 0.70 (0.68–0.71) | 0.76 (0.74–0.77) | 0.73 (0.71–0.74) |
| Somers' $D_{xy}$ | | 0.44 (0.41–0.48) | 0.57 (0.54–0.60) | 0.50 (0.46–0.54) |
| Threshold-level dichotomous $c$-index* | | 0.77 (0.75–0.78) | 0.82 (0.80–0.83) | 0.79 (0.78–0.80) |
| | GOSE > 1 | 0.83 (0.81–0.85) | 0.90 (0.88–0.92) | 0.86 (0.84–0.87) |
| | GOSE > 3 | 0.81 (0.79–0.83) | 0.86 (0.84–0.88) | 0.84 (0.83–0.86) |
| | GOSE > 4 | 0.78 (0.76–0.80) | 0.83 (0.80–0.85) | 0.82 (0.80–0.83) |
| | GOSE > 5 | 0.76 (0.74–0.77) | 0.80 (0.78–0.83) | 0.77 (0.75–0.79) |
| | GOSE > 6 | 0.72 (0.70–0.74) | 0.76 (0.73–0.79) | 0.75 (0.73–0.77) |
| | GOSE > 7 | 0.72 (0.69–0.74) | 0.75 (0.72–0.79) | 0.72 (0.70–0.75) |
| Threshold-level calibration slope* | | 0.98 (0.81–1.12) | 0.84 (0.76–0.91) | 1.00 (0.78–1.14) |
| | GOSE > 1 | 0.95 (0.78–1.10) | 0.98 (0.86–1.10) | 0.98 (0.78–1.14) |
| | GOSE > 3 | 0.97 (0.80–1.12) | 0.90 (0.80–1.02) | 1.05 (0.81–1.20) |
| | GOSE > 4 | 1.06 (0.86–1.23) | 0.89 (0.79–1.00) | 1.10 (0.85–1.27) |
| | GOSE > 5 | 1.01 (0.78–1.21) | 0.82 (0.72–0.94) | 1.01 (0.76–1.22) |
| | GOSE > 6 | 0.98 (0.73–1.20) | 0.74 (0.62–0.87) | 0.97 (0.70–1.20) |
| | GOSE > 7 | 0.92 (0.69–1.18) | 0.68 (0.54–0.83) | 0.89 (0.61–1.18) |

Data represent mean (95% confidence interval) for the best-performing model, per predictor set, based on a given metric. For threshold-level metrics, a single best-performing model, per predictor set, was determined by the overall unweighted average across the thresholds. Interpretations for each metric are provided in **Materials and methods**. Mean and confidence interval values were derived using bias-corrected bootstrapping (1,000 resamples) and represent the variation across repeated $k$-fold cross-validation folds (20 repeats of 5 folds) and, for the concise-predictor-based model (CPM) and the extended concise-predictor-based model (eCPM), 100 missing value imputations. CPM_Best = CPM with best value for given metric (**S2 Table**). APM_Best = all-predictor-based model (APM) with best value for given metric (**S3 Table**). eCPM_Best = eCPM with best value for given metric (**S4 Table**). GOSE = Glasgow Outcome Scale–Extended at 6 months post-injury.

*Values in these rows correspond to the unweighted average across all GOSE thresholds.

discrimination among the model types was observed in CPM_DeepOR, which had a significantly lower ORC and Somers' $D_{xy}$ than the other models. The discrimination performance metrics for each APM are listed in **S3 Table**. APM_MN had a significantly higher ORC, Somers' $D_{xy}$, and dichotomous $c$-indices at lower GOSE thresholds (i.e., GOSE > 1 and GOSE > 3) than did APM_OR. Moreover, in **S4 Appendix**, we see that the best-performing parametric configurations of APM_MN did not contain additional hidden layers between the token embedding and output layers. Our results of performance within predictor sets consistently demonstrate that increasing analytical complexity, in terms of using deep learning (for CPMs) or adding hidden network layers (for APMs), did not improve discrimination of outcomes. In the case of deep learning models, multinomial outcome encoding significantly outperformed ordinal outcome encoding (**Fig 1A**).

The discrimination performance metrics of the best-performing CPMs (CPM_Best), compared with those of the best-performing APMs (APM_Best), are listed in **Table 4**. In contrast to the case of analytical complexity, we observe that expanding the predictor set yielded a significant improvement in ORC, Somers' $D_{xy}$, and each threshold-level dichotomous $c$-index except for those of the highest GOSE thresholds (i.e., GOSE > 6 and GOSE > 7). On average, models trained on the concise predictor set (CPMs) correctly separated two randomly selected patients from two randomly selected GOSE categories 70% (95% CI: 68%– 71%) of the time, while models trained on all baseline predictors (APMs) in the CENTER-TBI dataset did so 76% (95% CI: 74%– 77%) of the time. These percentages also correspond to the average proportional closeness of predicted rankings to true GOSE rankings of patient sets. CPM_Best

explained 44% (95% CI: 41%– 48%) of the ordinal variation in GOSE while $APM_{Best}$ explained 57% (95% CI: 54%– 60%) in their respective model outputs. At increasing GOSE thresholds, the dichotomous $c$-indices of $CPM_{Best}$ and $APM_{Best}$, as well as the gap between them, consistently decreased (**Table 4**). This signifies that predicting higher 6-month functional outcomes is more difficult than predicting lower 6-month functional outcomes. Moreover, the gains in discrimination earned from expanding the predictor set mostly come from improved performance at lower GOSE thresholds (i.e., predicting survival, return of consciousness, or recovery of functional independence).

## CPM and APM calibration performance

The calibration slopes and calibration curves for each CPM are displayed in **S2 Table** and **S3 Fig**, respectively. Both logistic regression CPMs ($CPM_{MNLR}$ and $CPM_{POLR}$) are significantly overfitted at the three highest GOSE thresholds (i.e., GOSE > 5, GOSE > 6, and GOSE > 7). The graphical calibration of $CPM_{DeepOR}$ was significantly worse than that of the other CPMs (**S3 Fig**). The calibration slopes and calibration curves for each APM are displayed in **S3 Table** and **S4 Fig**, respectively. $APM_{OR}$ is poorly calibrated at each threshold of GOSE. $APM_{MN}$ is significantly overfitted at the three highest GOSE thresholds (i.e., GOSE > 5, GOSE > 6, and GOSE > 7).

The calibration slopes and calibration curves for the best-calibrated CPMs ($CPM_{Best}$), compared against those for the best-calibrated APMs ($APM_{Best}$), are displayed in **Table 4** and **Fig 3**, respectively. Unlike $CPM_{Best}$, $APM_{Best}$ could not avoid significant overfitting at the three highest GOSE thresholds (i.e., GOSE > 5, GOSE > 6, and GOSE > 7). At these thresholds, we observe that the calibration curve of $APM_{Best}$ significantly veered off the diagonal line of ideal calibration for higher predicted probabilities. However, due to the relative infrequency of these predictions (comparative histograms in **Fig 3**), the ICI of $APM_{Best}$ is not significantly higher than that of $CPM_{Best}$. Our results suggest that $APM_{Best}$ requires more patients with higher functional outcomes, in both the training and validation sets, to mitigate overfitting [45].

## Predictor importance

Given that $APM_{MN}$ significantly outperforms $APM_{OR}$ in discrimination and calibration, we focus the assessment of predictor importance to $APM_{MN}$. A bar plot of the mean absolute SHAP values associated with the 15 most important predictors in $APM_{MN}$ is provided in **Fig 4**. We find that the subjective early prognoses of ICU physicians had the greatest contribution towards $APM_{MN}$ predictions, particularly for the prediction of death (GOSE = 1) within 6 months. Initially, this result (along with the high contribution of other physician impressions) seems to suggest that integration of a physician's interpretations of a patient's baseline status may add important prognostic information. These impressions likely summarise information from a variable number of other predictors along with the physician's own experience-based judgement, resulting in high prediction contributions. However, inclusion of these variables may result in problematic self-fulfilling prophecies [51]. For instance, a physician's poor prognosis directly influences WLSM, which was instituted in 144 (70.2%) of the 205 patients who died in the ICU [52]. Including a variable for physician prognosis may then negatively bias the outcome prediction and unduly promote WLSM. Therefore, we do not consider physician impression predictors for our extended concise predictor set. We also observe that 'age at admission' was the only concise predictor among the 15 most important ones. The importance ranks (out of 1,151) of the concise predictors (**Table 2**) are: age = 5th, glucose = 23rd, Marshall CT = 25th, pupillary reactivity = 29th, GCSm = 42nd, haemoglobin = 50th, hypoxia = 284th, tSAH = 301st, EDH = 414th, and hypotension = 420th. The eight remaining predictors of the top 15 (**Fig 4**) were added to the concise predictor set to form our extended concise predictor

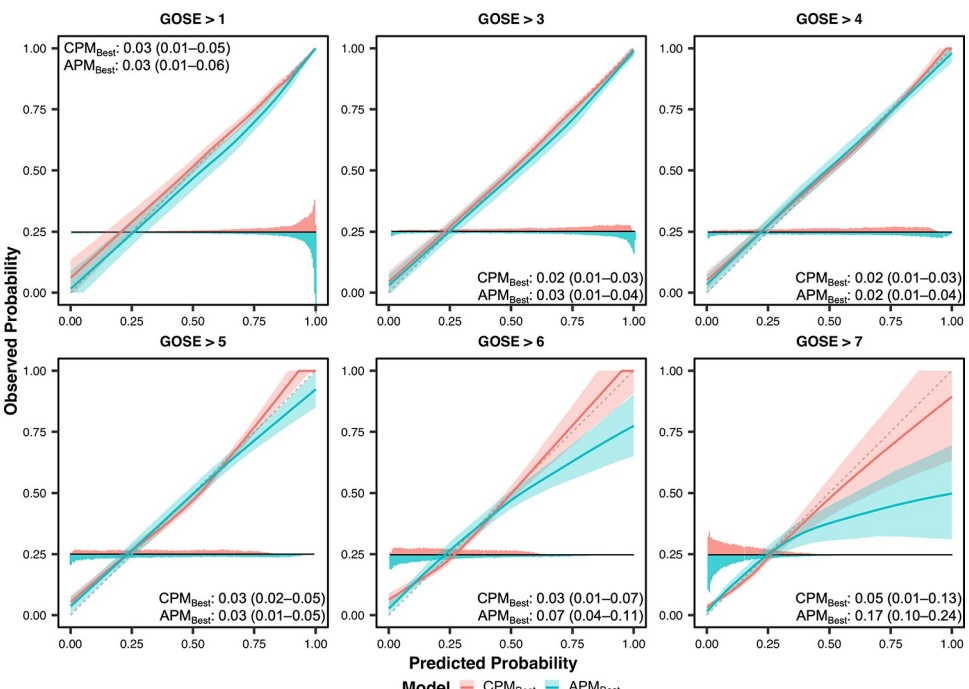

**Fig 3. Ordinal calibration curves of best-performing concise-predictor-based model (CPM$_{Best}$) and best-performing all-predictor-based model (APM$_{Best}$).** GOSE = Glasgow Outcome Scale–Extended at 6 months post-injury. In each panel, a comparative histogram (200 uniform bins), centred at a horizontal line in the bottom quarter, displays the distribution of predicted probabilities for CPM$_{Best}$ (above the line) and APM$_{Best}$ (below the line) at the given GOSE threshold. CPM$_{Best}$ and APM$_{Best}$ correspond to the CPM (**S2 Table**) and APM (**S3 Table**), respectively, with the lowest unweighted average of integrated calibration indices (ICI) across the thresholds. Shaded areas are 95% confidence intervals derived using bias-corrected bootstrapping (1,000 resamples) to represent the variation across repeated *k*-fold cross-validation folds (20 repeats of 5 folds) and, for CPM$_{Best}$, 100 missing value imputations. The values in each panel correspond to the mean ICI (95% confidence interval) at the given threshold. The diagonal dashed line represents the line of perfect calibration (ICI = 0).

set. Within the tokens for "employment status before injury," we found that the single token indicating retirement is much more important than the others. Thus, instead of encoding all 10 options for employment status, we included a single indicator variable for retirement in our extended concise predictor set. The eight added predictors included 2 demographic variables (retirement status and highest level of formal education), 4 protein biomarker concentrations (neurofilament light chain [NFL], glial fibrillary acidic protein [GFAP], total tau protein [T-tau], and S100 calcium-binding protein B [S100B]), and 2 clinical assessment variables (worst abbreviated injury score [AIS] among head, neck, brain, and cervical spine injuries and incidence of post-traumatic amnesia at ICU admission). The extended concise predictor set, including the ten original concise predictors and the eight added predictors, is statistically characterised in **S1 Table**.

A bar plot of the mean absolute SHAP values of APM$_{MN}$ for each of the five folds of the first repeat is provided in **S5 Fig**. Most of the eight added predictors, along with age at admission, are consistently represented among the most important predictors across the five folds.

## eCPM discrimination and calibration

The discrimination and calibration metrics for the best-performing extended-predictor-based model (eCPM$_{Best}$) are listed in **Table 4**. Inclusion of the eight selected predictors accounted for about half of the gains in discrimination performance achieved by APM$_{Best}$ over CPM$_{Best}$

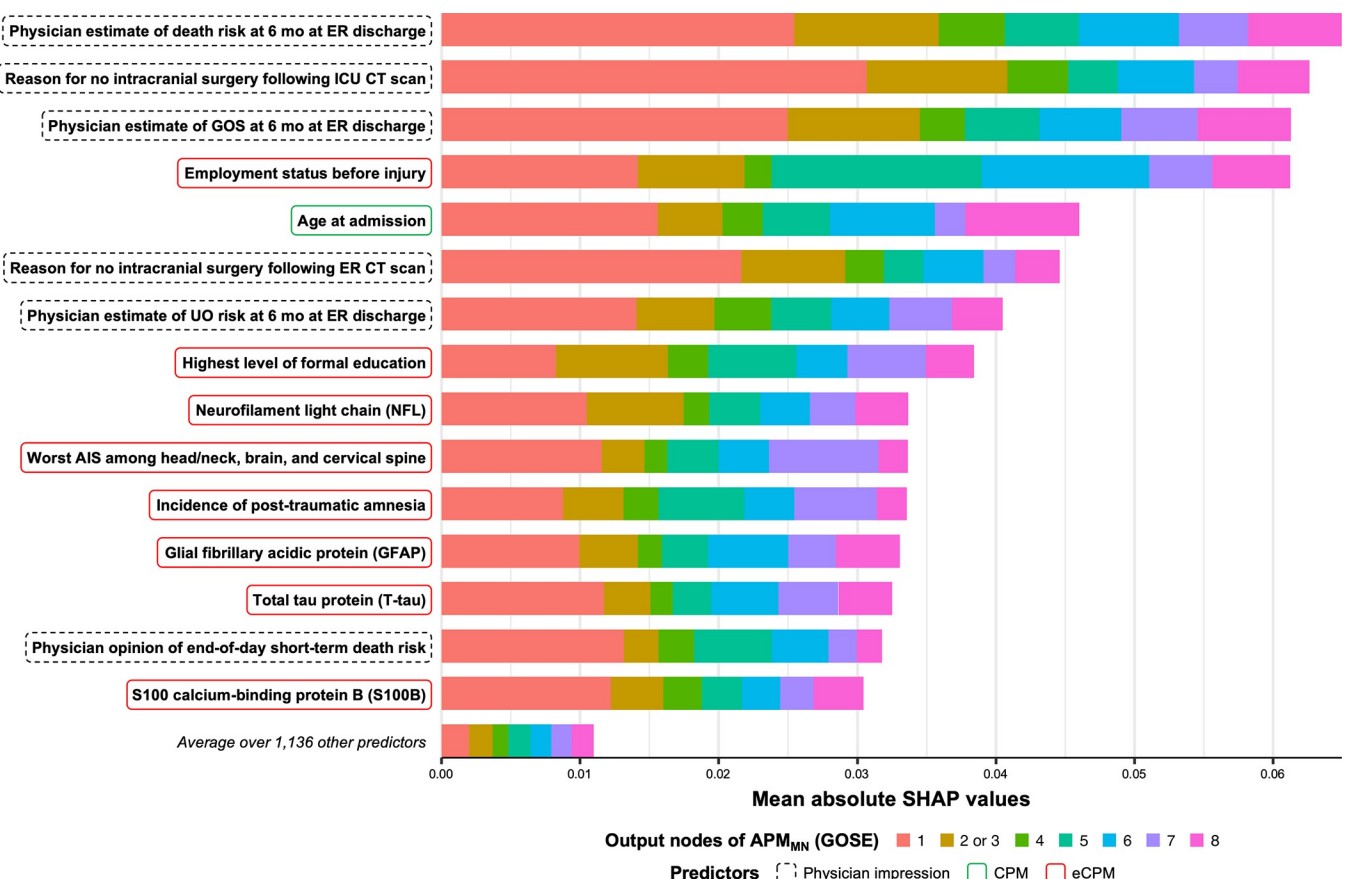

**Fig 4. Mean absolute shapley additive explanation (SHAP) values of most important predictors for multinomial-encoding all-predictor-based model (APM$_{MN}$).** ICU = intensive care unit. ER = emergency room. CT = computerised tomography. GOS = Glasgow Outcome Scale (not extended). UO = unfavourable outcome, defined by functional dependence (i.e., GOSE ≤ 4). AIS = Abbreviated Injury Scale. GOSE = Glasgow Outcome Scale–Extended at 6 months post-injury. CPM = predictors that are included in the original concise predictor set. eCPM = predictors that are added to the original concise predictor set to form the extended concise predictor set. The mean absolute SHAP value is interpreted as the average magnitude of the relative additive contribution of a predictor's most important token towards the predicted probability at each GOSE score for a single patient. Predictor types are denoted by the coloured boundary around predictor names. Physician impression predictors denote predictors that encode the explicit impressions or rationales of ICU physicians and are not considered for the extended concise predictor set.

according to ORC, Somers' $D_{xy}$, and the dichotomous *c*-indices. Based on the difference in Somers' $D_{xy}$, the eight added predictors allowed models to explain an additional 6% of the ordinal variation in GOSE at 6 months post-injury. Unlike APM$_{Best}$, eCPM$_{Best}$ is not significantly overfitted at any threshold. The calibration curves of eCPMs (**S6 Fig**) are largely similar to those of the corresponding CPMs (**S3 Fig**), except at the highest threshold (i.e., GOSE > 7). Similar to those of APM$_{MN}$, the calibration curves of eCPMs veer off the line of ideal calibration at higher predicted probabilities of GOSE > 7. The eCPM results support the finding that discrimination performance can be improved with the expansion of the predictor set. Furthermore, by limiting the number of added predictors and the analytical complexity of the model, eCPM avoided the significant miscalibration of APM at higher thresholds.

The discrimination and calibration metrics for each eCPM are listed in **S4 Table**.

## Discussion

To our knowledge, this is the most comprehensive evaluation of early ordinal outcome prognosis for critically ill TBI patients. Our analysis cross-compares a range of ordinal prediction

modelling strategies with a large range of available baseline predictors to determine the relative contribution of each towards model performance. Employing an AI tokenisation and embedding technique, we develop highly flexible ordinal prediction models that can learn from the entire, heterogeneous set of 1,151 predictors, available within the first 24 hours of ICU stay, in the CENTER-TBI dataset. This information includes not only all baseline clinical data currently deemed significant for ICU care of TBI but also advanced sub-study results (e.g., protein biomarkers, central haemostatic markers, genetic markers, and advanced MRI results) that represent the experimental frontier of clinical TBI assessment [1, 15, 16]. Therefore, our work reveals the interpretable limits of baseline ordinal, 6-month GOSE prediction in the ICU at this time.

Our key finding is that augmenting the baseline predictor set was much more relevant for improving ordinal model prediction performance than was increasing analytical complexity with deep learning. Within a given predictor set, artificial neural networks did not perform better than logistic regression models (**S2 and S4 Tables**), nor did models with additional hidden layers for the APMs (**S4 Appendix**). This result is consistent with findings in the binary prediction case [53]. On the other hand, augmenting the predictor set, from CPM to APM, substantially improved ordinal discrimination (ORC: +8.6%, **Table 4**) and prediction at lower GOSE thresholds (e.g., GOSE > 1 $c$-index: +8.4%, **Table 4**). Just adding eight predictors to the concise predictor set accounted for about half of the gains in discrimination. However, the addition of predictors negatively affected model calibration, particularly at higher GOSE thresholds (**Fig 3**, **Table 4**). This result underlines the need for careful consideration of probability calibration during model development (e.g., recalibrate with isotonic regression to mitigate overfitting).

At the same time, our results also indicate that ordinal early outcome prognosis for critically ill TBI patients is limited in capability. The best-performing model, which learns from all baseline information in the CENTER-TBI dataset, can only correctly discriminate two randomly chosen patients with two randomly chosen GOSE scores 76% (95% CI: 74%– 77%) of the time. Equivalently, if the best performing model was tasked with ranking seven randomly chosen patients–each with a different true GOSE–by predicted GOSE, an average 5.10 (95% CI: 4.74– 5.46) of the 21 possible pairwise orderings will be incorrect. Currently, ordinal model outputs explain, at best, 57% (95% CI: 54%– 60%) of the ordinal variation in 6-month GOSE. Ordinal prediction models struggle to reliably predict full recovery (GOSE > 7 $c$-index: 75% [95% CI: 72%– 79%], **Table 4**), and gains from expanding the predictor set diminish with higher GOSE thresholds.

It is important to acknowledge that the predictor importance results of this article should not be interpreted for predictor discovery or validation. SHAP values are visualised (**Fig 4**) solely to globally interpret $APM_{MN}$ predictions and to form the extended concise predictor set. Risk factor validation, which falls out of the scope of this work, would require investigating the robustness and clinical plausibility of the relationship between predictor values and their corresponding SHAP values [54]. Moreover, causal analysis with apt consideration of confounding factors or dataset biases would be necessary before commenting on the potential effects or mechanisms of individual predictors.

We recognise several limitations in our study. While the concise predictor set was originally designed for prognosis after moderate-to-severe TBI [8] (i.e., baseline GCS 3–12), 26.6% of our study population had experienced mild (i.e., baseline GCS 13–15) TBI (**Table 1**). Predictor sets have been designed for mild TBI patients (e.g., UPFRONT study predictors [55]). However, in line with the aims of the CENTER-TBI project [15], we focus the TBI population not by initial characterisation with GCS but by stratum of care (i.e., admission to the ICU). Therefore, we selected the single concise predictor set that was best validated for the majority of critically ill TBI patients. Our outcome categories (GOSE at 6 months post-injury) were statistically imputed for 13% of our dataset using available GOSE between 2 weeks and one-

year post-injury. Although this method was strongly validated on the same (CENTER-TBI) dataset [18], we do recognise that our outcome labels may not be precisely correct. The focus of this work is on the prediction of functional outcomes through GOSE; nonetheless, it is worth considering other outcomes, such as quality-of-life and psychological health, that are important for clinical decision making [56]. Finally, before the AI models developed in this work and in subsequent iterations could be integrated into ICU practice, limitations of generalisability must be addressed [57]. Our models were developed on a multicentre, adult population, prospectively recruited between 2014 and 2017 [25], across Europe, and may encode recruitment, collection, and clinical biases native to our patient set. AI models must continuously be updated, iteratively retrained on incoming information, to help fight the effect these biases may have on returned prognoses for a given patient.

In the setting of TBI prognosis, we encourage the use of AI not to add analytical complexity (i.e., make models "deeper") but to expand the predictor set (i.e., make models "wider"). Studies have uncovered promising prognostic value in neuro-inflammatory markers [58, 59] and high-resolution TBI monitoring and imaging modalities (e.g., intracranial and cerebral perfusion pressure [60–62], accelerometery [63], and MRI [64–66]), and we recommend integrating these features into ordinal prognostic models, especially to improve prediction of higher functional outcomes. We also believe that there is a feasible performance limit to reliable ordinal outcome prognosis if only statically considering the clinical information from the first 24 hours of ICU stay. It would seem far-fetched to expect all relevant information pertaining to an outcome at 6 months to be encapsulated in the first 24 hours of ICU treatment. Heterogeneous pathophysiological processes unfold over time in patients after TBI [67, 68], and dynamic prediction models, which return model outputs longitudinally with changing clinical information, are better equipped to consider these temporal effects on prognosis. Dynamic prognosis models have been developed for TBI patients [69] and the greater ICU population (not exclusive to TBI) [35, 70, 71], but none of them predict functional outcomes on an ordinal scale. We suggest that the next iteration of this work should be to develop ordinal dynamic prediction models on all clinical information available during the complete ICU stay.

## Ethical approval statement

The CENTER-TBI study has been conducted in accordance with all relevant laws of the European Union and all relevant laws of the country where the recruiting sites were located, including (but not limited to) the relevant privacy and data protection laws and regulations, the relevant laws and regulations on the use of human materials, and all relevant guidance relating to clinical studies from time in force including (but not limited to) the ICH Harmonised Tripartite Guideline for Good Clinical Practice (CPMP/ICH/135/95) and the World Medical Association Declaration of Helsinki entitled "Ethical Principles for Medical Research Involving Human Subjects." Written informed consent by the patients and/or the legal representative/next of kin was obtained (according to local legislation) for all patients recruited in the core dataset of CENTER-TBI and documented in the electronic case report form. Ethical approval was obtained for each recruiting site.

The list of sites, ethical committees, approval numbers and approval dates can be found on the website: https://www.center-tbi.eu/project/ethical-approval.

## Supporting information

**S1 Appendix. Explanation of selected ordinal prediction models for CPM and eCPM.**
(PDF)

**S2 Appendix. Explanation of APM for ordinal GOSE prediction.**
(PDF)

**S3 Appendix. Detailed explanation of ordinal model performance and calibration metrics.**
(PDF)

**S4 Appendix. Hyperparameter optimisation results.**
(PDF)

**S1 Fig. CONSORT-style flow diagram for patient enrolment and follow-up.**
CENTER-TBI = Collaborative European NeuroTrauma Effectiveness Research in TBI.
ICU = intensive care unit. GOSE = Glasgow Outcome Scale–Extended. MSM = Markov multi-state model (see **Materials and methods**). The dashed, olive-green line in the lower-middle of the diagram divides the enrolment flow diagram (above) and the follow-up breakdown (below).
(TIF)

**S2 Fig. Characterisation of missingness among concise predictor set.** U.P. = unreactive pupils. GCSm = motor component score of the Glasgow Coma Scale. Hb = haemoglobin. Glu. = glucose. HoTN = hypotension. Marshall = Marshall computerised tomography classification. tSAH = traumatic subarachnoid haemorrhage. EDH = extradural haematoma. (**A**) Proportion of total sample size ($n$ = 1,550) with missing values for each IMPACT extended model predictor. (**B**) Missingness matrix where each column represents a concise predictor, and each row represents a combination of missing predictors (red) and non-missing predictors (blue) found in the dataset. The prevalence of each combination (i.e., row) in the study population is shown with a horizontal histogram (far right) labelled with the proportion of the study population with the corresponding combination of missing predictors. For example, the bottom row of the matrix shows that 54.77% of the study population had no missing concise predictors while the penultimate row shows that 14.71% of the study population had only glucose and haemoglobin missing among the concise predictors.
(TIF)

**S3 Fig. Ordinal calibration curves of each concise-predictor-based model (CPM).**
GOSE = Glasgow Outcome Scale–Extended at 6 months post-injury. Shaded areas are 95% confidence intervals derived using bias-corrected bootstrapping (1,000 resamples) to represent the variation across repeated $k$-fold cross-validation folds (20 repeats of 5 folds) and 100 missing value imputations. The values in each panel correspond to the mean integrated calibration index (ICI) (95% confidence interval) at the given threshold. The diagonal dashed line represents the line of perfect calibration (ICI = 0). The CPM types ($CPM_{MNLR}$, $CPM_{POLR}$, $CPM_{DeepMN}$, and $CPM_{DeepOR}$) are decoded in the **Materials and methods** and described in **S1 Appendix**.
(TIF)

**S4 Fig. Ordinal calibration curves of each all-predictor-based model (APM).**
GOSE = Glasgow Outcome Scale–Extended at 6 months post-injury. Shaded areas are 95% confidence intervals derived using bias-corrected bootstrapping (1,000 resamples) to represent the variation across repeated $k$-fold cross-validation folds (20 repeats of 5 folds). The values in each panel correspond to the mean integrated calibration index (ICI) (95% confidence interval) at the given threshold. The diagonal dashed line represents the line of perfect calibration (ICI = 0). The APM types ($APM_{MN}$ and $APM_{OR}$) are decoded in the **Materials and methods** and described in **S2 Appendix**.
(TIF)

**S5 Fig. Mean absolute SHAP values of the most important predictors for APM$_{MN}$ in each of the five folds of the first repeat.** ICU = intensive care unit. CT = computerised tomography. ER = emergency room. GOS = Glasgow Outcome Scale (not extended). AIS = Abbreviated Injury Scale. UO = unfavourable outcome, defined by functional dependence (i.e., GOSE $\leq$ 4). FIBTEM = fibrin-based extrinsically activated test with tissue factor and cytochalasin D. GOSE = Glasgow Outcome Scale–Extended at 6 months post-injury. The mean absolute SHAP value is interpreted as the average magnitude of the relative additive contribution of a predictor's most important token towards the predicted probability at each GOSE score for a single patient.
(TIF)

**S6 Fig. Ordinal calibration curves of each extended concise-predictor-based model (eCPM).** GOSE = Glasgow Outcome Scale–Extended at 6 months post-injury. Shaded areas are 95% confidence intervals derived using bias-corrected bootstrapping (1,000 resamples) to represent the variation across repeated *k*-fold cross-validation folds (20 repeats of 5 folds) and 100 missing value imputations. The values in each panel correspond to the mean integrated calibration index (ICI) (95% confidence interval) at the given threshold. The diagonal dashed line represents the line of perfect calibration (ICI = 0). The eCPM types (eCPM$_{MNLR}$, eCPM$_{POLR}$, eCPM$_{DeepMN}$, and eCPM$_{DeepOR}$) are decoded in the **Materials and methods** and described in **S1 Appendix**.
(TIF)

**S1 Table. Extended concise baseline predictors of the study population stratified by ordinal 6-month outcomes.**
(PDF)

**S2 Table. Ordinal concise-predictor-based model (CPM) discrimination and calibration performance.**
(PDF)

**S3 Table. Ordinal all-predictor-based model (APM) discrimination and calibration performance.**
(PDF)

**S4 Table. Ordinal extended concise-predictor-based model (eCPM) discrimination and calibration performance.**
(PDF)

## Acknowledgments

We are grateful to the patients and families of our study for making our efforts to improve TBI care and outcome possible.

S.B. would like to thank: Abhishek Dixit (Univ. of Cambridge) for helping access the CENTER-TBI dataset, Jacob Deasy (Univ. of Cambridge) for aiding the development of modelling methodology, and Kathleen Mitchell-Fox (Princeton Univ.) for offering comments on the manuscript. All authors would like to thank Andrew I. R. Maas (Antwerp Univ. Hospital) for offering comments on the manuscript.

## The CENTER-TBI investigators and participants

The co-lead investigators of CENTER-TBI are designated with an asterisk (*), and their contact email addresses are listed below.

Cecilia Åkerlund[1], Krisztina Amrein[2], Nada Andelic[3], Lasse Andreassen[4], Audny Anke[5], Anna Antoni[6], Gérard Audibert[7], Philippe Azouvi[8], Maria Luisa Azzolini[9], Ronald Bartels[10], Pál Barzó[11], Romuald Beauvais[12], Ronny Beer[13], Bo-Michael Bellander[14], Antonio Belli[15], Habib Benali[16], Maurizio Berardino[17], Luigi Beretta[9], Morten Blaabjerg[18], Peter Bragge[19], Alexandra Brazinova[20], Vibeke Brinck[21], Joanne Brooker[22], Camilla Brorsson[23], Andras Buki[24], Monika Bullinger[25], Manuel Cabeleira[26], Alessio Caccioppola[27], Emiliana Calappi[27], Maria Rosa Calvi[9], Peter Cameron[28], Guillermo Carbayo Lozano[29], Marco Carbonara[27], Simona Cavallo[17], Giorgio Chevallard[30], Arturo Chieregato[30], Giuseppe Citerio[31,32], Hans Clusmann[33], Mark Coburn[34], Jonathan Coles[35], Jamie D. Cooper[36], Marta Correia[37], Amra Čović [38], Nicola Curry[39], Endre Czeiter[24], Marek Czosnyka[26], Claire Dahyot-Fizelier[40], Paul Dark[41], Helen Dawes[42], Véronique De Keyser[43], Vincent Degos[16], Francesco Della Corte[44], Hugo den Boogert[10], Bart Depreitere[45], Đula Đilvesi[46], Abhishek Dixit[47], Emma Donoghue[22], Jens Dreier[48], Guy-Loup Dulière[49], Ari Ercole[47], Patrick Esser[42], Erzsébet Ezer[50], Martin Fabricius[51], Valery L. Feigin[52], Kelly Foks[53], Shirin Frisvold[54], Alex Furmanov[55], Pablo Gagliardo[56], Damien Galanaud[16], Dashiell Gantner[28], Guoyi Gao[57], Pradeep George[58], Alexandre Ghuysen[59], Lelde Giga[60], Ben Glocker[61], Jagoš Golubovic[46], Pedro A. Gomez[62], Johannes Gratz[63], Benjamin Gravesteijn[64], Francesca Grossi[44], Russell L. Gruen[65], Deepak Gupta[66], Juanita A. Haagsma[64], Iain Haitsma[67], Raimund Helbok[13], Eirik Helseth[68], Lindsay Horton[69], Jilske Huijben[64], Peter J. Hutchinson[70], Bram Jacobs[71], Stefan Jankowski[72], Mike Jarrett[21], Ji-yao Jiang[58], Faye Johnson[73], Kelly Jones[52], Mladen Karan[46], Angelos G. Kolias[70], Erwin Kompanje[74], Daniel Kondziella[51], Evgenios Kornaropoulos[47], Lars-Owe Koskinen[75], Noémi Kovács[76], Ana Kowark[77], Alfonso Lagares[62], Linda Lanyon[58], Steven Laureys[78], Fiona Lecky[79,80], Didier Ledoux[78], Rolf Lefering[81], Valerie Legrand[82], Aurelie Lejeune[83], Leon Levi[84], Roger Lightfoot[85], Hester Lingsma[64], Andrew I.R. Maas[43,*], Ana M. Castaño-León[62], Marc Maegele[86], Marek Majdan[20], Alex Manara[87], Geoffrey Manley[88], Costanza Martino[89], Hugues Maréchal[49], Julia Mattern[90], Catherine McMahon[91], Béla Melegh[92], David Menon[47,*], Tomas Menovsky[43], Ana Mikolic[64], Benoit Misset[78], Visakh Muraleedharan[58], Lynnette Murray[28], Ancuta Negru[93], David Nelson[1], Virginia Newcombe[47], Daan Nieboer[64], József Nyirádi[2], Otesile Olubukola[79], Matej Oresic[94], Fabrizio Ortolano[27], Aarno Palotie[95,96,97], Paul M. Parizel[98], Jean-François Payen[99], Natascha Perera[12], Vincent Perlbarg[16], Paolo Persona[100], Wilco Peul[101], Anna Piippo-Karjalainen[102], Matti Pirinen[95], Dana Pisica[64], Horia Ples[93], Suzanne Polinder[64], Inigo Pomposo[29], Jussi P. Posti[103], Louis Puybasset[104], Andreea Radoi[105], Arminas Ragauskas[106], Rahul Raj[102], Malinka Rambadagalla[107], Isabel Retel Helmrich[64], Jonathan Rhodes[108], Sylvia Richardson[109], Sophie Richter[47], Samuli Ripatti[95], Saulius Rocka[106], Cecilie Roe[110], Olav Roise[111,112], Jonathan Rosand[113], Jeffrey V. Rosenfeld[114], Christina Rosenlund[115], Guy Rosenthal[55], Rolf Rossaint[77], Sandra Rossi[100], Daniel Rueckert[61] Martin Rusnák[116], Juan Sahuquillo[105], Oliver Sakowitz[90,117], Renan Sanchez-Porras[117], Janos Sandor[118], Nadine Schäfer[81], Silke Schmidt[119], Herbert Schoechl[120], Guus Schoonman[121], Rico Frederik Schou[122], Elisabeth Schwendenwein[6], Charlie Sewalt[64], Ranjit D. Singh[101], Toril Skandsen[123,124,] Peter Smielewski[26], Abayomi Sorinola[125], Emmanuel Stamatakis[47], Simon Stanworth[39], Robert Stevens[126], William Stewart[127], Ewout W. Steyerberg[64,128], Nino Stocchetti[129], Nina Sundström[130], Riikka Takala[131], Viktória Tamás[125], Tomas Tamosuitis[132], Mark Steven Taylor[20], Braden Te Ao[52], Olli Tenovuo[103], Alice Theadom[52], Matt Thomas[87], Dick Tibboel[133], Marjolein Timmers[74], Christos Tolias[134], Tony Trapani[28], Cristina Maria Tudora[93], Andreas Unterberg[90], Peter Vajkoczy [135], Shirley Vallance[28], Egils Valeinis[60], Zoltán Vámos[50], Mathieu van der Jagt[136], Gregory Van der Steen[43], Joukje van der Naalt[71], Jeroen T.J.M. van Dijck[101], Inge A. M. van Erp[101], Thomas A. van Essen[101], Wim Van Hecke[137], Caroline van Heugten[138], Dominique Van Praag[139], Ernest van Veen[64], Thijs Vande Vyvere[137], Roel P. J. van Wijk[101], Alessia Vargiolu[32], Emmanuel Vega[83], Kimberley Velt[64], Jan Verheyden[137], Paul M. Vespa[140],

Anne Vik[123,141], Rimantas Vilcinis[132], Victor Volovici[67], Nicole von Steinbüchel[38], Daphne Voormolen[64], Petar Vulekovic[46], Kevin K.W. Wang[142], Daniel Whitehouse[47], Eveline Wiegers[64], Guy Williams[47], Lindsay Wilson[69], Stefan Winzeck[47], Stefan Wolf[143], Zhihui Yang[113], Peter Ylén[144], Alexander Younsi[90], Frederick A. Zeiler[47,145], Veronika Zelinkova[20], Agate Ziverte[60,] Tommaso Zoerle[27]

[1] Department of Physiology and Pharmacology, Section of Perioperative Medicine and Intensive Care, Karolinska Institutet, Stockholm, Sweden

[2] János Szentágothai Research Centre, University of Pécs, Pécs, Hungary

[3] Division of Surgery and Clinical Neuroscience, Department of Physical Medicine and Rehabilitation, Oslo University Hospital and University of Oslo, Oslo, Norway

[4] Department of Neurosurgery, University Hospital Northern Norway, Tromso, Norway

[5] Department of Physical Medicine and Rehabilitation, University Hospital Northern Norway, Tromso, Norway

[6] Trauma Surgery, Medical University Vienna, Vienna, Austria

[7] Department of Anesthesiology & Intensive Care, University Hospital Nancy, Nancy, France

[8] Raymond Poincare hospital, Assistance Publique–Hopitaux de Paris, Paris, France

[9] Department of Anesthesiology & Intensive Care, S Raffaele University Hospital, Milan, Italy

[10] Department of Neurosurgery, Radboud University Medical Center, Nijmegen, The Netherlands

[11] Department of Neurosurgery, University of Szeged, Szeged, Hungary

[12] International Projects Management, ARTTIC, Munchen, Germany

[13] Department of Neurology, Neurological Intensive Care Unit, Medical University of Innsbruck, Innsbruck, Austria

[14] Department of Neurosurgery & Anesthesia & intensive care medicine, Karolinska University Hospital, Stockholm, Sweden

[15] NIHR Surgical Reconstruction and Microbiology Research Centre, Birmingham, UK

[16] Anesthesie-Réanimation, Assistance Publique–Hopitaux de Paris, Paris, France

[17] Department of Anesthesia & ICU, AOU Città della Salute e della Scienza di Torino—Orthopedic and Trauma Center, Torino, Italy

[18] Department of Neurology, Odense University Hospital, Odense, Denmark

[19] BehaviourWorks Australia, Monash Sustainability Institute, Monash University, Victoria, Australia

[20] Department of Public Health, Faculty of Health Sciences and Social Work, Trnava University, Trnava, Slovakia

[21] Quesgen Systems Inc., Burlingame, California, USA

[22] Australian & New Zealand Intensive Care Research Centre, Department of Epidemiology and Preventive Medicine, School of Public Health and Preventive Medicine, Monash University, Melbourne, Australia

[23] Department of Surgery and Perioperative Science, Umeå University, Umeå, Sweden

[24] Department of Neurosurgery, Medical School, University of Pécs, Hungary and Neurotrauma Research Group, János Szentágothai Research Centre, University of Pécs, Hungary

[25] Department of Medical Psychology, Universitätsklinikum Hamburg-Eppendorf, Hamburg, Germany

[26] Brain Physics Lab, Division of Neurosurgery, Dept of Clinical Neurosciences, University of Cambridge, Addenbrooke's Hospital, Cambridge, UK

[27] Neuro ICU, Fondazione IRCCS Cà Granda Ospedale Maggiore Policlinico, Milan, Italy

[28]ANZIC Research Centre, Monash University, Department of Epidemiology and Preventive Medicine, Melbourne, Victoria, Australia

[29]Department of Neurosurgery, Hospital of Cruces, Bilbao, Spain

[30]NeuroIntensive Care, Niguarda Hospital, Milan, Italy

[31]School of Medicine and Surgery, Università Milano Bicocca, Milano, Italy

[32]NeuroIntensive Care, ASST di Monza, Monza, Italy

[33]Department of Neurosurgery, Medical Faculty RWTH Aachen University, Aachen, Germany

[34]Department of Anesthesiology and Intensive Care Medicine, University Hospital Bonn, Bonn, Germany

[35]Department of Anesthesia & Neurointensive Care, Cambridge University Hospital NHS Foundation Trust, Cambridge, UK

[36]School of Public Health & PM, Monash University and The Alfred Hospital, Melbourne, Victoria, Australia

[37]Radiology/MRI department, MRC Cognition and Brain Sciences Unit, Cambridge, UK

[38]Institute of Medical Psychology and Medical Sociology, Universitätsmedizin Göttingen, Göttingen, Germany

[39]Oxford University Hospitals NHS Trust, Oxford, UK

[40]Intensive Care Unit, CHU Poitiers, Potiers, France

[41]University of Manchester NIHR Biomedical Research Centre, Critical Care Directorate, Salford Royal Hospital NHS Foundation Trust, Salford, UK

[42]Movement Science Group, Faculty of Health and Life Sciences, Oxford Brookes University, Oxford, UK

[43]Department of Neurosurgery, Antwerp University Hospital and University of Antwerp, Edegem, Belgium

[44]Department of Anesthesia & Intensive Care, Maggiore Della Carità Hospital, Novara, Italy

[45]Department of Neurosurgery, University Hospitals Leuven, Leuven, Belgium

[46]Department of Neurosurgery, Clinical centre of Vojvodina, Faculty of Medicine, University of Novi Sad, Novi Sad, Serbia

[47]Division of Anaesthesia, University of Cambridge, Addenbrooke's Hospital, Cambridge, UK

[48]Center for Stroke Research Berlin, Charité –Universitätsmedizin Berlin, corporate member of Freie Universität Berlin, Humboldt-Universität zu Berlin, and Berlin Institute of Health, Berlin, Germany

[49]Intensive Care Unit, CHR Citadelle, Liège, Belgium

[50]Department of Anaesthesiology and Intensive Therapy, University of Pécs, Pécs, Hungary

[51]Departments of Neurology, Clinical Neurophysiology and Neuroanesthesiology, Region Hovedstaden Rigshospitalet, Copenhagen, Denmark

[52]National Institute for Stroke and Applied Neurosciences, Faculty of Health and Environmental Studies, Auckland University of Technology, Auckland, New Zealand

[53]Department of Neurology, Erasmus MC, Rotterdam, the Netherlands

[54]Department of Anesthesiology and Intensive care, University Hospital Northern Norway, Tromso, Norway

[55]Department of Neurosurgery, Hadassah-hebrew University Medical center, Jerusalem, Israel

[56]Fundación Instituto Valenciano de Neurorrehabilitación (FIVAN), Valencia, Spain

[57]Department of Neurosurgery, Shanghai Renji hospital, Shanghai Jiaotong University/school of medicine, Shanghai, China

[58]Karolinska Institutet, INCF International Neuroinformatics Coordinating Facility, Stockholm, Sweden

[59]Emergency Department, CHU, Liège, Belgium

[60]Neurosurgery clinic, Pauls Stradins Clinical University Hospital, Riga, Latvia

[61]Department of Computing, Imperial College London, London, UK

[62]Department of Neurosurgery, Hospital Universitario 12 de Octubre, Madrid, Spain

[63]Department of Anesthesia, Critical Care and Pain Medicine, Medical University of Vienna, Austria

[64]Department of Public Health, Erasmus Medical Center-University Medical Center, Rotterdam, The Netherlands

[65]College of Health and Medicine, Australian National University, Canberra, Australia

[66]Department of Neurosurgery, Neurosciences Centre & JPN Apex trauma centre, All India Institute of Medical Sciences, New Delhi-110029, India

[67]Department of Neurosurgery, Erasmus MC, Rotterdam, the Netherlands

[68]Department of Neurosurgery, Oslo University Hospital, Oslo, Norway

[69]Division of Psychology, University of Stirling, Stirling, UK

[70]Division of Neurosurgery, Department of Clinical Neurosciences, Addenbrooke's Hospital & University of Cambridge, Cambridge, UK

[71]Department of Neurology, University of Groningen, University Medical Center Groningen, Groningen, Netherlands

[72]Neurointensive Care, Sheffield Teaching Hospitals NHS Foundation Trust, Sheffield, UK

[73]Salford Royal Hospital NHS Foundation Trust Acute Research Delivery Team, Salford, UK

[74]Department of Intensive Care and Department of Ethics and Philosophy of Medicine, Erasmus Medical Center, Rotterdam, The Netherlands

[75]Department of Clinical Neuroscience, Neurosurgery, Umeå University, Umeå, Sweden

[76]Hungarian Brain Research Program—Grant No. KTIA_13_NAP-A-II/8, University of Pécs, Pécs, Hungary

[77]Department of Anaesthesiology, University Hospital of Aachen, Aachen, Germany

[78]Cyclotron Research Center, University of Liège, Liège, Belgium

[79]Centre for Urgent and Emergency Care Research (CURE), Health Services Research Section, School of Health and Related Research (ScHARR), University of Sheffield, Sheffield, UK

[80]Emergency Department, Salford Royal Hospital, Salford UK

[81]Institute of Research in Operative Medicine (IFOM), Witten/Herdecke University, Cologne, Germany

[82]VP Global Project Management CNS, ICON, Paris, France

[83]Department of Anesthesiology-Intensive Care, Lille University Hospital, Lille, France

[84]Department of Neurosurgery, Rambam Medical Center, Haifa, Israel

[85]Department of Anesthesiology & Intensive Care, University Hospitals Southhampton NHS Trust, Southhampton, UK

[86]Cologne-Merheim Medical Center (CMMC), Department of Traumatology, Orthopedic Surgery and Sportmedicine, Witten/Herdecke University, Cologne, Germany

[87]Intensive Care Unit, Southmead Hospital, Bristol, Bristol, UK

[88]Department of Neurological Surgery, University of California, San Francisco, California, USA

[89]Department of Anesthesia & Intensive Care,M. Bufalini Hospital, Cesena, Italy

[90]Department of Neurosurgery, University Hospital Heidelberg, Heidelberg, Germany

[91]Department of Neurosurgery, The Walton centre NHS Foundation Trust, Liverpool, UK

[92]Department of Medical Genetics, University of Pécs, Pécs, Hungary

[93]Department of Neurosurgery, Emergency County Hospital Timisoara, Timisoara, Romania

[94]School of Medical Sciences, Örebro University, Örebro, Sweden

[95]Institute for Molecular Medicine Finland, University of Helsinki, Helsinki, Finland

[96]Analytic and Translational Genetics Unit, Department of Medicine; Psychiatric & Neuro-developmental Genetics Unit, Department of Psychiatry; Department of Neurology, Massachusetts General Hospital, Boston, MA, USA

[97]Program in Medical and Population Genetics; The Stanley Center for Psychiatric Research, The Broad Institute of MIT and Harvard, Cambridge, MA, USA

[98]Department of Radiology, University of Antwerp, Edegem, Belgium

[99]Department of Anesthesiology & Intensive Care, University Hospital of Grenoble, Grenoble, France

[100]Department of Anesthesia & Intensive Care, Azienda Ospedaliera Università di Padova, Padova, Italy

[101]Dept. of Neurosurgery, Leiden University Medical Center, Leiden, The Netherlands and Dept. of Neurosurgery, Medical Center Haaglanden, The Hague, The Netherlands

[102]Department of Neurosurgery, Helsinki University Central Hospital

[103]Division of Clinical Neurosciences, Department of Neurosurgery and Turku Brain Injury Centre, Turku University Hospital and University of Turku, Turku, Finland

[104]Department of Anesthesiology and Critical Care, Pitié -Salpêtrière Teaching Hospital, Assistance Publique, Hôpitaux de Paris and University Pierre et Marie Curie, Paris, France

[105]Neurotraumatology and Neurosurgery Research Unit (UNINN), Vall d'Hebron Research Institute, Barcelona, Spain

[106]Department of Neurosurgery, Kaunas University of technology and Vilnius University, Vilnius, Lithuania

[107]Department of Neurosurgery, Rezekne Hospital, Latvia

[108]Department of Anaesthesia, Critical Care & Pain Medicine NHS Lothian & University of Edinburg, Edinburgh, UK

[109]Director, MRC Biostatistics Unit, Cambridge Institute of Public Health, Cambridge, UK

[110]Department of Physical Medicine and Rehabilitation, Oslo University Hospital/University of Oslo, Oslo, Norway

[111]Division of Orthopedics, Oslo University Hospital, Oslo, Norway

[112]Institue of Clinical Medicine, Faculty of Medicine, University of Oslo, Oslo, Norway

[113]Broad Institute, Cambridge MA Harvard Medical School, Boston MA, Massachusetts General Hospital, Boston MA, USA

[114]National Trauma Research Institute, The Alfred Hospital, Monash University, Melbourne, Victoria, Australia

[115]Department of Neurosurgery, Odense University Hospital, Odense, Denmark

[116]International Neurotrauma Research Organisation, Vienna, Austria

[117]Klinik für Neurochirurgie, Klinikum Ludwigsburg, Ludwigsburg, Germany

[118]Division of Biostatistics and Epidemiology, Department of Preventive Medicine, University of Debrecen, Debrecen, Hungary

[119]Department Health and Prevention, University Greifswald, Greifswald, Germany

[120]Department of Anaesthesiology and Intensive Care, AUVA Trauma Hospital, Salzburg, Austria

[121]Department of Neurology, Elisabeth-TweeSteden Ziekenhuis, Tilburg, the Netherlands

[122]Department of Neuroanesthesia and Neurointensive Care, Odense University Hospital, Odense, Denmark

[123]Department of Neuromedicine and Movement Science, Norwegian University of Science and Technology, NTNU, Trondheim, Norway

[124]Department of Physical Medicine and Rehabilitation, St.Olavs Hospital, Trondheim University Hospital, Trondheim, Norway

[125]Department of Neurosurgery, University of Pécs, Pécs, Hungary

[126]Division of Neuroscience Critical Care, Johns Hopkins University School of Medicine, Baltimore, USA

[127]Department of Neuropathology, Queen Elizabeth University Hospital and University of Glasgow, Glasgow, UK

[128]Dept. of Department of Biomedical Data Sciences, Leiden University Medical Center, Leiden, The Netherlands

[129]Department of Pathophysiology and Transplantation, Milan University, and Neuroscience ICU, Fondazione IRCCS Cà Granda Ospedale Maggiore Policlinico, Milano, Italy

[130]Department of Radiation Sciences, Biomedical Engineering, Umeå University, Umeå, Sweden

[131]Perioperative Services, Intensive Care Medicine and Pain Management, Turku University Hospital and University of Turku, Turku, Finland

[132]Department of Neurosurgery, Kaunas University of Health Sciences, Kaunas, Lithuania

[133]Intensive Care and Department of Pediatric Surgery, Erasmus Medical Center, Sophia Children's Hospital, Rotterdam, The Netherlands

[134]Department of Neurosurgery, Kings college London, London, UK

[135]Neurologie, Neurochirurgie und Psychiatrie, Charité –Universitätsmedizin Berlin, Berlin, Germany

[136]Department of Intensive Care Adults, Erasmus MC–University Medical Center Rotterdam, Rotterdam, the Netherlands

[137]icoMetrix NV, Leuven, Belgium

[138]Movement Science Group, Faculty of Health and Life Sciences, Oxford Brookes University, Oxford, UK

[139]Psychology Department, Antwerp University Hospital, Edegem, Belgium

[140]Director of Neurocritical Care, University of California, Los Angeles, USA

[141]Department of Neurosurgery, St.Olavs Hospital, Trondheim University Hospital, Trondheim, Norway

[142]Department of Emergency Medicine, University of Florida, Gainesville, Florida, USA

[143]Department of Neurosurgery, Charité –Universitätsmedizin Berlin, corporate member of Freie Universität Berlin, Humboldt-Universität zu Berlin, and Berlin Institute of Health, Berlin, Germany

[144]VTT Technical Research Centre, Tampere, Finland

[145]Section of Neurosurgery, Department of Surgery, Rady Faculty of Health Sciences, University of Manitoba, Winnipeg, MB, Canada

*Co-lead investigators: andrew.maas@uza.be (AIRM) and dkm13@cam.ac.uk (DM)

## Author Contributions

**Conceptualization:** Shubhayu Bhattacharyay, Ari Ercole.

**Data curation:** Shubhayu Bhattacharyay.

**Formal analysis:** Shubhayu Bhattacharyay, Ioan Milosevic, Ari Ercole.

**Funding acquisition:** Shubhayu Bhattacharyay, David K. Menon, Ari Ercole.

**Investigation:** Shubhayu Bhattacharyay, Ari Ercole.

**Methodology:** Shubhayu Bhattacharyay, Ioan Milosevic, Ewout W. Steyerberg, David W. Nelson, Ari Ercole.

**Project administration:** Shubhayu Bhattacharyay, David W. Nelson, Ari Ercole.

**Resources:** Shubhayu Bhattacharyay, David K. Menon, Ari Ercole.

**Software:** Shubhayu Bhattacharyay, Ioan Milosevic.

**Supervision:** David K. Menon, Ewout W. Steyerberg, David W. Nelson, Ari Ercole.

**Validation:** Shubhayu Bhattacharyay.

**Visualization:** Shubhayu Bhattacharyay.

**Writing – original draft:** Shubhayu Bhattacharyay.

**Writing – review & editing:** Shubhayu Bhattacharyay, Ioan Milosevic, Lindsay Wilson, David K. Menon, Robert D. Stevens, Ewout W. Steyerberg, David W. Nelson, Ari Ercole.

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
