## [Decision Letter · Decision Letter 0]

24 Apr 2022

PONE-D-22-05175The leap to ordinal: functional prognosis after traumatic brain injury using artificial intelligencePLOS ONE

Dear Dr. Bhattacharyay,

Thank you for submitting your manuscript to PLOS ONE. After careful consideration, we feel that it has merit but does not fully meet PLOS ONE’s publication criteria as it currently stands. Therefore, we invite you to submit a revised version of the manuscript that addresses the points raised during the review process.

We look forward to receiving your revised manuscript.

Kind regards,

Soojin Park, M.D.

Academic Editor

PLOS ONE

Journal Requirements:

(The research was supported by the National Institute for Health Research (NIHR) Brain Injury MedTech Co-operative based at Cambridge University Hospitals NHS Foundation Trust and University of Cambridge. The views expressed are those of the author(s) and not necessarily those of the NHS, NIHR or the Department of Health and Social Care.

CENTER-TBI was supported by the European Union 7th Framework programme (EC grant 602150). Additional funding was obtained from the Hannelore Kohl Stiftung (Germany), from OneMind (USA), and from Integra LifeSciences Corporation (USA).

CSD3 is supported by the United Kingdom Engineering and Physical Sciences Research Council (EPSRC Tier-2 capital grant EP/T022159/1).

SB is currently funded by a Gates Cambridge fellowship. 

The funders had no role in study design, data collection and analysis, decision to publish, or preparation of the manuscript.)

(The research was supported by the National Institute for Health Research (NIHR) Brain Injury MedTech Co-operative based at Cambridge University Hospitals NHS Foundation Trust and University of Cambridge. The views expressed are those of the author(s) and not necessarily those of the NHS, NIHR or the Department of Health and Social Care.

CENTER-TBI was supported by the European Union 7th Framework programme (EC grant 602150). Additional funding was obtained from the Hannelore Kohl Stiftung (Germany), from OneMind (USA), and from Integra LifeSciences Corporation (USA). We are grateful to the patients of our study for helping us in our efforts to improve TBI care and outcome. We gratefully acknowledge interactions and support from the International Initiative for TBI Research (InTBIR) investigators.

CSD3 is supported by the United Kingdom Engineering and Physical Sciences Research Council (EPSRC Tier-2 capital grant EP/T022159/1).

S.B. is currently funded by a Gates Cambridge fellowship. S.B. would like to thank: Abhishek Dixit (Univ. of Cambridge) for helping access the CENTER-TBI dataset, Jacob Deasy (Univ. of Cambridge) for aiding the development of modelling methodology, and Kathleen Mitchell-Fox (Princeton Univ.) for offering comments on the manuscript. All authors would like to thank Andrew I. R. Maas (Antwerp Univ. Hospital) for offering comments on the manuscript.)

(The research was supported by the National Institute for Health Research (NIHR) Brain Injury MedTech Co-operative based at Cambridge University Hospitals NHS Foundation Trust and University of Cambridge. The views expressed are those of the author(s) and not necessarily those of the NHS, NIHR or the Department of Health and Social Care.

CENTER-TBI was supported by the European Union 7th Framework programme (EC grant 602150). Additional funding was obtained from the Hannelore Kohl Stiftung (Germany), from OneMind (USA), and from Integra LifeSciences Corporation (USA).

CSD3 is supported by the United Kingdom Engineering and Physical Sciences Research Council (EPSRC Tier-2 capital grant EP/T022159/1).

6. One of the noted authors is a group or consortium CENTER-TBI investigators and participants. In addition to naming the author group, please list the individual authors and affiliations within this group in the acknowledgments section of your manuscript. Please also indicate clearly a lead author for this group along with a contact email address.

Reviewers' comments:

Reviewer's Responses to Questions

**Comments to the Author**

1. Is the manuscript technically sound, and do the data support the conclusions?

Reviewer #1: Yes

Reviewer #2: Partly

2. Has the statistical analysis been performed appropriately and rigorously? 

Reviewer #1: Yes

Reviewer #2: Yes

3. Have the authors made all data underlying the findings in their manuscript fully available?

Reviewer #1: Yes

Reviewer #2: Yes

4. Is the manuscript presented in an intelligible fashion and written in standard English?

Reviewer #1: Yes

Reviewer #2: Yes

5. Review Comments to the Author

Reviewer #1: This is a novel manuscript which can add significantly to the body of our knowledge in TBI management. I recommend acceptance with minor revisions.

I have a few suggestions below, which I hope authors can consider to improve their work.

Title: as the title explicitly mentions “artificial intelligence”, it would be great if the authors could add a few sentences in the introduction to expand on the importance of artificial intelligence in TBI and hence make some connections to similar work done in the field. Alternatively- which I think would be a more suitable proposition- authors can substitute “artificial intelligence” with “clinical predictive model” in the title.

Line 76: “Ethically…” There are extensive ethical debates on using AI in medicine and autonomy of patients; this sentence is not in the context here and does not help the flow of the text. I would suggest the authors to remove it or expand it in a separate paragraph.

Line 86: “Without …” This sentence is quite vague. Please rephrase it.

Line 122: “However, …” I believe an additional challenge would be that the current predictive model designed would not function in a different data set. For example, in paediatric TBI patients below the age of 16 who are excluded in the study. Please elaborate on this.

Line 288: “… categorical predictors” Could the authors please elaborate the categorical predictors they used here. This was not clear. Was it based on the physician?

Line 539: “We find that …” Do authors believe this is a strength or caveat to their predictive model? Please elaborate.

Line 553: “The eight remaining …” I do not suggest the authors to collect and re-analyse the data, but can they elaborate if they considered including any inflammatory markers in their predictive model? Please justify in the text.

Line 560: “tau protein” Can the author explain if they consider any connection between the presence of cognitive decline, Alzheimer disease and TBI in their model?

Line 631: “This means” Please rephrase this sentence. It is not very clear.

Line 675: “greater ICU population” Please clarify it that you mean non-TBI ICU patients.

Reviewer #2: This work answers an important question to predict the functional outcome of TBI patients on an 8-point GOSE scale rather than dichotomized GOSE on threshold 4 using ordinal classification models.

Major comment:

- How do you assure that extended features that are brought in supplementary table 1 do not enforce historical biases? For example, it is surprising that “Highest formal education” is picked as a predictor. In a rather simplistic analysis, patients with primary school education tend to suffer from proportionally worse outcome (GOSE 1/GOSE 8 = 31/19 = 1.63) compared to patients with “University degree” (GOSE 1/GOSE 8 = 26/35 = 0.74). Thus, this feature can be a proxy for a patient’ wealth status and the level of care they received. These features, although increase the classification performance in a retrospective study, might not be clinically meaningful and so not applicable in real clinical settings.

- In supplementary table 1, can being “retired” be a proxy of and highly correlated with age, and not a risk factor by itself? Please comment on this.

- Please explain more on how the missing GOSE at 5 to 8-month labels were imputed using data available at 2-week to 1-year post-injury? And wouldn’t removing those cases be more preferable than adding estimation noise to the labels, especially for the ones that label was generated using GOSE at 2-month? Please provide the summary statistics of the recorded GOSE for these cases with missing GOSE 5 to 8-month.

- The fold-wise average SHAP value is an ad-hoc method for evaluating the overall SHAP contribution. A related publication on GOSE prediction [1] showed that SHAP contributions can be non-robust across different runs. For example, in Figure 1 of [1], the authors show contribution of creatinine can vary from -0.02 to 0.015 in one experiment and vary from -0.06 to 0.01 in another with different behaviors. Please comment on this non-robustness of SHAP values and in addition to overall SHAP contribution plots in Figure 4, provide the SHAP contribution plots for each fold separately.

[1] Farzaneh, Negar, Craig A. Williamson, Jonathan Gryak, and Kayvan Najarian. "A hierarchical expert-guided machine learning framework for clinical decision support systems: an application to traumatic brain injury prognostication." NPJ digital medicine 4, no. 1 (2021): 1-9.

- In Figure 4, explain how are “physician estimate of UO risk at 6 mo at ER discharge” and “physician estimate of GOS at 6 mo at ER discharge” among the predictors while this feature is not available within the 24-hr post-admission? It was mentioned that patients were excluded if discharged before 24-hr, so all patients stayed at ICU for at least 24 hr post-admission, thus this parameter is not supposed to be gathered before the 24-hr period.

- Following on the previous comment, are all subjective physician impression features (including “Physician estimate of death risk at 6 mo post injury”, “Reason for no intracranial surgery following CT scan”, “Physician estimate of GOS at 6 mo at ER discharge”, “Reason for no intracranial surgery following ER CT scan”, “Physician estimate of UO risk at ER discharge”, “Physician opinion of end-of-day short-term death risk”) always collected during the first 24-hr post-admission?

Minor comments:

- It is possible that the discriminant features between GOSE 1 and 2 are different from discriminant features between GOSE 7 and 8. So using a same pool of features for different thresholds in a single model to discriminate between all 8 points might not take advantage of the full potential of all discriminating feature. Please provide the performance of predicting p(GOSE>1), …p (GOSE>7) using 6 binary classifiers, each trained on fixed thresholds of 1, 3, 4, 5, 6, 7 and compare its results to the ordinal classifier’s performance.

- Supplementary Figure 2 B is not easily understandable. Add more info on how to interpret the figure.

- In Figure 4, does “Reason for no intracranial surgery following CT scan” also include “Reason for no intracranial surgery following ER CT scan”? or it means “Reason for no intracranial surgery following outside-ER CT scan”?

6. PLOS authors have the option to publish the peer review history of their article (what does this mean?). If published, this will include your full peer review and any attached files.

Reviewer #1: No

Reviewer #2: No

---

## [Author Response · Author response to Decision Letter 0]

3 May 2022

Specific Responses: 

Response to Reviewer #1:

Comment 1: This is a novel manuscript which can add significantly to the body of our knowledge in TBI management. I recommend acceptance with minor revisions. I have a few suggestions below, which I hope authors can consider to improve their work.

Reply 1: Thank you for your time in reviewing our work and for your insightful comments.

Comment 2: Title: as the title explicitly mentions “artificial intelligence”, it would be great if the authors could add a few sentences in the introduction to expand on the importance of artificial intelligence in TBI and hence make some connections to similar work done in the field. Alternatively- which I think would be a more suitable proposition- authors can substitute “artificial intelligence” with “clinical predictive model” in the title.

Reply 2: We agree with your proposition to change the title. To highlight the novel modelling techniques employed in this work, we have decided to replace “artificial intelligence” with “a flexible modelling approach.” We have also added “detailed” before “functional prognosis.” We hope these two points help distinguish this work from prior studies that have developed binary clinical predictive models with conventional statistical methods. The full amended title is “The leap to ordinal: detailed functional prognosis after traumatic brain injury with a flexible modelling approach.”

Comment 3: Line 76: “Ethically…” There are extensive ethical debates on using AI in medicine and autonomy of patients; this sentence is not in the context here and does not help the flow of the text. I would suggest the authors to remove it or expand it in a separate paragraph.

Reply 3: We have restructured the beginning of this paragraph to support the narrative flow and removed the mention of ethics as suggested ([line 76-79]). We believe the revised text better highlights the critical flaw of dichotomised prediction: it imposes a universal prediction threshold of GOSE, thereby limiting individual choice of favourability when an empirically supported ideal threshold does not exist. 

Comment 4: Line 86: “Without …” This sentence is quite vague. Please rephrase it.

Reply 4: We agree. We have rephrased the sentence to a clearer if-then statement, followed by an example ([line 87-89]).

Comment 5: Line 122: “However, …” I believe an additional challenge would be that the current predictive model designed would not function in a different data set. For example, in paediatric TBI patients below the age of 16 who are excluded in the study. Please elaborate on this.

Reply 5: We agree that limited transferability is an important limitation of our work. Since it is not a challenge specific to ordinal prediction models, which we discuss in this section of the Introduction, we have elaborated your point in the limitations section of the Discussion ([lines 670-676]). 

Comment 6: Line 288: “… categorical predictors” Could the authors please elaborate the categorical predictors they used here. This was not clear. Was it based on the physician?

Reply 6: We apologize for the unclear language here. We simply mean that, after removing all formatting from text entries to free-form predictors (e.g., physician comments), we append the unformatted text to the predictor name. To avoid further confusion, we have modified the text here to just mention the concatenation of unformatted text and the predictor name ([line 287-289]).

Comment 7: Line 539: “We find that …” Do authors believe this is a strength or caveat to their predictive model? Please elaborate.

Reply 7: We have added a few sentences [lines 541-551] to the article to comment on this important point. On one hand, it is interesting to observe the model identify physician estimates as important predictors. It shows that the model recognised a direct predictor of the outcome while underlining the potential impact of physician integration/cooperation with these models. On the other hand, physician prognoses are a potentially problematic predictor. The withdrawal of life-sustaining measures (WLSM) is a direct result of a poor prognosis, and we acknowledge that inclusion of this predictor in clinical prediction models may result in self-fulfilling prophecies [R1]. For example, a poor initial prognosis from a physician may negatively bias the model’s prediction of outcome and unduly promote WLSM. Therefore, we crucially do not include this predictor, or other physician impressions, in the extended predictor set. We believe that the improvement observed in the extended predictor set (over the concise predictor set) without these subjective variables is an overall achievement for the modelling approach.

Comment 8: Line 553: “The eight remaining …” I do not suggest the authors to collect and re-analyse the data, but can they elaborate if they considered including any inflammatory markers in their predictive model? Please justify in the text.

Reply 8: Thank you for this interesting question. In the CENTER-TBI study, specific neuro-inflammatory markers (i.e., cytokines) were only analysed in a very limited and selected subset of the population – too few to permit including in the analysis. We do have routine hospital lab reports from most patients in the study, but the potential inflammatory markers are limited to CRP, WBC, and Neutrophil/Lymphocyte ratios. Given the association of cytokines with outcome, we have added a mention to neuroinflammatory markers in the Discussion ([line 680-681]) as an additional set of predictors to investigate.

Comment 9: Line 560: “tau protein” Can the author explain if they consider any connection between the presence of cognitive decline, Alzheimer disease and TBI in their model?

Reply 9: While potentially interesting, this is unfortunately outside of the scope of this article. Our objective was to validate and interpret the models for ordinal prediction and not the predictors themselves. We demonstrate the most important predictors primarily to understand how the model made its predictions. Therefore, we kindly wish to avoid making claims on the predictors without rigorous validation. We have elaborated this point in an added paragraph ([lines 647-654]) to the Discussion. 

Comment 10: Line 631: “This means” Please rephrase this sentence. It is not very clear.

Reply 10: We apologise for the obscurity of our original sentence. We have rephrased the sentence in clearer language ([lines 639-642]).

Comment 11: Line 675: “greater ICU population” Please clarify it that you mean non-TBI ICU patients.

Reply 11: Thank you for this point. These ICU models do also include TBI patients, so instead of “non-TBI,” we have added “(not exclusive to TBI)” ([line 692-693]).

Response to Reviewer #2:

Comment 1: This work answers an important question to predict the functional outcome of TBI patients on an 8-point GOSE scale rather than dichotomized GOSE on threshold 4 using ordinal classification models.

Reply 1: Thank you for your time in reviewing our work and for your insightful comments.

Comment 2: Major comment: How do you assure that extended features that are brought in supplementary table 1 do not enforce historical biases? For example, it is surprising that “Highest formal education” is picked as a predictor. In a rather simplistic analysis, patients with primary school education tend to suffer from proportionally worse outcome (GOSE 1/GOSE 8 = 31/19 = 1.63) compared to patients with “University degree” (GOSE 1/GOSE 8 = 26/35 = 0.74). Thus, this feature can be a proxy for a patient’ wealth status and the level of care they received. These features, although increase the classification performance in a retrospective study, might not be clinically meaningful and so not applicable in real clinical settings.

Reply 2: Thank you for this important comment about the possible enforcement of historical biases in our selection of extended predictors. We acknowledge in the Discussion ([lines 670-676]) that AI models are highly susceptible to dataset bias. Moreover, in [lines 647-654], we explicitly mention that our objective was to validate and understand the performance limits of ordinal prediction models and not to validate specific predictors themselves. Therefore, we also disclose that our predictor importance results should be interpreted not for predictor discovery/validation but rather for model interpretation ([lines 647-650]). In terms of highest level of education, we entirely acknowledge the confound that you propose, and we integrate an acknowledgment about confounding factors in [line 653]. However, we believe there are two points worth considering. First, socioeconomic variables of patients and their families are available in the CENTER-TBI database (CRF: https://center-tbi.incf.org/static/pdf/DemographicsandSocioeconomicStatus.pdf). Thus, if wealth status was mostly responsible for the high explanatory power of “highest level of education,” we would expect to see other socioeconomic variables (e.g., job category, living situation, or parents’ background) ranked higher than level of education. Second, level of education is a key indicator of cognitive reserve. Cognitive reserve (often measured through IQ, level of education, and National Adult Reading Test [NART]) is an independently validated predictor of functional recovery (through neural adaptability) from TBI [R2-6]. Therefore, we believe that the inclusion of education level in our extended predictor model is not completely unwarranted given that we account for other socioeconomic factors and that our objective is to simply test the limits of performance in ordinal prediction models.

Comment 3: In supplementary table 1, can being “retired” be a proxy of and highly correlated with age, and not a risk factor by itself? Please comment on this.

Reply 3: While we agree that retirement status is strongly correlated with age, the multivariate analysis performed with SHAP would not identify retirement status as a more important predictor than age if retirement status had no explanatory power outside of its correlation with age [R7]. Moreover, a value for age is not missing for a single patient in the dataset (S2 Fig), while employment status is missing for 238 patients (15.35% of study population) (S1 Table). Therefore, SHAP would more likely identify age, the more available predictor in the dataset, as the more important predictor overall if retirement status did not add any information over its correlation with age [R7]. At the same time, we acknowledge that the high predictor importance ascribed to retirement status may be caused by another confounding factor not explained in the predictor set or an inherent bias of the dataset. Before considering retirement status as an independent and understandable risk factor, one would have to perform rigorous predictor validation, which, as we mention in the previous reply and in [line 650], is outside of the scope of this article.

Comment 4: Please explain more on how the missing GOSE at 5 to 8-month labels were imputed using data available at 2-week to 1-year post-injury? And wouldn’t removing those cases be more preferable than adding estimation noise to the labels, especially for the ones that label was generated using GOSE at 2-month? Please provide the summary statistics of the recorded GOSE for these cases with missing GOSE 5 to 8-month.

Reply 4: Statistically, reliable imputation of missing clinical outcomes is preferable to complete case analysis (i.e., removing all patients with missing GOSE) when one cannot certainly claim that the data is missing completely at random (MCAR) [R8-12]. Especially in the case of follow-up clinical assessments, when there are likely significant factors that lead to missingness [R13], complete case analysis would, in comparison to validated imputation methods, introduce bias, decrease statistical power, and underestimate the width of confidence intervals [R9-12]. In our case, as mentioned in [lines 172-174], we impute GOSE with a Markov multi-state model (MSM) that was validated on (and calibrated to) the same dataset by Kunzmann et al. [R14] It is important to note that this approach uses trajectory analysis, estimating 6-month GOSE from all available between 2 weeks and 1-year post-injury per patient. This contrasts with methods such as “Last Observation Carried Forward” (LOCF), where imputation is based on a single GOSE. Not only does the MSM more accurately and reliably estimate missing GOSE than does LOCF [R14] but also a trajectory method using multiple datapoints per patient mitigates the risk (of lower outcomes) that would result from only using very early assessments. The summary statistics of the recorded GOSE for the imputed cases can be found in Fig 1 of the article by Kunzmann et al. [R14]. This figure shows the distribution of observed GOSEs at the three other follow-up timepoints (2 weeks, 3 months, and 12 months post-injury) that were used for imputation. We direct readers to this article with a citation in [line 174].

Comment 5: The fold-wise average SHAP value is an ad-hoc method for evaluating the overall SHAP contribution. A related publication on GOSE prediction [1] showed that SHAP contributions can be non-robust across different runs. For example, in Figure 1 of [1], the authors show contribution of creatinine can vary from -0.02 to 0.015 in one experiment and vary from -0.06 to 0.01 in another with different behaviors. Please comment on this non-robustness of SHAP values and in addition to overall SHAP contribution plots in Figure 4, provide the SHAP contribution plots for each fold separately. [1] Farzaneh, Negar, Craig A. Williamson, Jonathan Gryak, and Kayvan Najarian. "A hierarchical expert-guided machine learning framework for clinical decision support systems: an application to traumatic brain injury prognostication." NPJ digital medicine 4, no. 1 (2021): 1-9.

Reply 5: Given that we have 100 partitions (20 repeats of 5 folds), it would be difficult to visualise the SHAP contribution of each partition in Fig 4, which already visualises 3 dimensions and long predictor names. Therefore, we have added a separate figure (S5 Fig) which visualises the SHAP contributions of the most import features for each of the 5 folds in the first repeat. We have also added a mention to this supplementary figure in the Results ([lines 582-585]). Regarding the article by Farzaneh et al. [R15], it is important to note that the authors investigate the non-robustness of the directionality of the relationship between predictor and SHAP values (via Kendall’s tau) and not the non-robustness of SHAP magnitude. Like the magnitude bar plots in Supplementary Figure 2 in [R15], our magnitude bar plots (Fig 4 and S5 Fig) would not necessarily be able to capture non-robust behaviour of predictor-SHAP directionality. The directionality of predictor-SHAP relationships is certainly important for global predictor validation. However, the objective of our article is the validation of ordinal prediction models for TBI and not the validation of predictors nor their potential relationship with GOSE. Inspired your comment, we have added a paragraph to the Discussion ([lines 647-654]) which emphasises this point and urges readers to read our SHAP results not as predictor validation but as model interpretation.

Comment 6: In Figure 4, explain how are “physician estimate of UO risk at 6 mo at ER discharge” and “physician estimate of GOS at 6 mo at ER discharge” among the predictors while this feature is not available within the 24-hr post-admission? It was mentioned that patients were excluded if discharged before 24-hr, so all patients stayed at ICU for at least 24 hr post-admission, thus this parameter is not supposed to be gathered before the 24-hr period.

Reply 6: We apologise for the miscommunication here. The prognosis was performed by physicians at emergency room (ER) discharge before ICU admission. The patients for whom this variable is available were discharged from the ER and then admitted to the ICU. Hence, this information would be available before the 24-hour post-ICU-admission cut-off.

Comment 7: Following on the previous comment, are all subjective physician impression features (including “Physician estimate of death risk at 6 mo post injury”, “Reason for no intracranial surgery following CT scan”, “Physician estimate of GOS at 6 mo at ER discharge”, “Reason for no intracranial surgery following ER CT scan”, “Physician estimate of UO risk at ER discharge”, “Physician opinion of end-of-day short-term death risk”) always collected during the first 24-hr post-admission?

Reply 7: Each of these variables are timestamped in the CENTER-TBI database (CRF: https://center-tbi.incf.org/static/pdf/ERTherapyanddischarge.pdf and https://center-tbi.incf.org/static/pdf/ImagingCTMRI.pdf). These variables were only included in token set of a patient if the timestamps fell within the first 24 hours of ICU stay. Therefore, even if the variables were not collected in the first 24 hours for all patients, they were included in our study only for the patients in the times they did. This process is described (for all predictors) in the “Design of all-predictor-based models (APMs)” ([lines 257-334]) section of the Methods. 

Comment 8: Minor comments: It is possible that the discriminant features between GOSE 1 and 2 are different from discriminant features between GOSE 7 and 8. So using a same pool of features for different thresholds in a single model to discriminate between all 8 points might not take advantage of the full potential of all discriminating feature. Please provide the performance of predicting p(GOSE>1), …p (GOSE>7) using 6 binary classifiers, each trained on fixed thresholds of 1, 3, 4, 5, 6, 7 and compare its results to the ordinal classifier’s performance.

Reply 8: We agree that different combinations of features are likely to be discriminative for different thresholds of GOSE. To clarify, each of the outcome encoding strategies for ordinal prediction (Fig 1A) allow for the model to flexibly learn different patterns of discriminant features for different scores and thresholds of the GOSE. Therefore, in Fig 4, we examine predictor importance for different GOSE scores in different colours. Even though our ordinal prediction models (of a specific predictor set) are trained on a single, all-encompassing pool of features, they are capable of learning different patterns of discrimination for different thresholds, which can be interpreted with SHAP. As we specify in [lines 85-91] of the Introduction, it is not appropriate to interpret independently trained and calibrated (i.e., unconstrained) models across the GOSE thresholds concurrently. This is because, without a constrained context of the other GOSE thresholds during training, a combination of prediction model outputs may be nonsensical. For example, Pr(GOSE > 1) may be less from one model than Pr(GOSE > 4) from another, even though both are independently calibrated. Therefore, we kindly believe that outputs from ordinal prediction models should not be compared with outputs from a set of independently trained binary prediction models.

Comment 9: Supplementary Figure 2 B is not easily understandable. Add more info on how to interpret the figure.

Reply 9: We have added text to the figure legend for S2B Fig ([lines 1336-1343]) to help clarify the missingness combination matrix and provide an example for interpretation.

Comment 10: In Figure 4, does “Reason for no intracranial surgery following CT scan” also include “Reason for no intracranial surgery following ER CT scan”? or it means “Reason for no intracranial surgery following outside-ER CT scan”?

Reply 10: “Reason for no intracranial surgery following CT scan” represents the latter: it comes from CT scans taken after ER discharge and after ICU admission. Since both “Reason for no intracranial surgery following CT scan” and “Reason for no intracranial surgery following ER CT scan” are included in Fig 4, we agree that it is worthwhile to further distinguish the two. We have added “ICU” to the former in Fig 4.

References

R1. Izzy S, Compton R, Carandang R, Hall W, Muehlschlegel S. Self-Fulfilling Prophecies Through Withdrawal of Care: Do They Exist in Traumatic Brain Injury, Too? Neurocrit Care. 2013;19: 347-363. doi: 10.1007/s12028-013-9925-z.

R2. Schneider EB, Sur S, Raymont V, Duckworth J, Kowalski RG, Efron DT, et al. Functional recovery after moderate/severe traumatic brain injury: A role for cognitive reserve? Neurology. 2014;82: 1636-1642. doi: 10.1212/WNL.0000000000000379.

R3. Fraser EE, Downing MG, Biernacki K, McKenzie DP, Ponsford JL. Cognitive Reserve and Age Predict Cognitive Recovery after Mild to Severe Traumatic Brain Injury. J Neurotrauma. 2019;36: 2753-2761. doi: 10.1089/neu.2019.6430.

R4. Nunnari D, Bramanti P, Marino S. Cognitive reserve in stroke and traumatic brain injury patients. Ital J Neurol Sci. 2014;35: 1513-1518. doi: 10.1007/s10072-014-1897-z.

R5. Steward KA, Kennedy R, Novack TA, Crowe M, Marson DC, Triebel KL. The Role of Cognitive Reserve in Recovery From Traumatic Brain Injury. J Head Trauma Rehabil. 2018;33: E18-E27. doi: 10.1097/HTR.0000000000000325.

R6. Donders J, Stout J. The Influence of Cognitive Reserve on Recovery from Traumatic Brain Injury. Arch Clin Neuropsychol. 2018;34: 206-213. doi: 10.1093/arclin/acy035.

R7. Lundberg SM, Lee S. A Unified Approach to Interpreting Model Predictions. In: Guyon I, Luxburg UV, Bengio S, Wallach H, Fergus R, Vishwanathan S, Garnett R, editors. Advances in Neural Information Processing Systems 30 (NIPS 2017). Long Beach: NIPS; 2017.

R8. Austin PC, White IR, Lee DS, van Buuren S. Missing Data in Clinical Research: A Tutorial on Multiple Imputation. Can J Cardiol. 2021;37: 1322-1331. doi: 10.1016/j.cjca.2020.11.010.

R9. van der Heijden, Geert J. M. G., T. Donders AR, Stijnen T, Moons KGM. Imputation of missing values is superior to complete case analysis and the missing-indicator method in multivariable diagnostic research: A clinical example. J Clin Epidemiol. 2006;59: 1102-1109. doi: 10.1016/j.jclinepi.2006.01.015.

R10. White IR, Carlin JB. Bias and efficiency of multiple imputation compared with complete-case analysis for missing covariate values. Stat Med. 2010;29: 2920-2931. doi: 10.1002/sim.3944.

R11. Ibrahim JG, Chu H, Chen M. Missing Data in Clinical Studies: Issues and Methods. J Clin Oncol. 2012;30: 3297-3303. doi: 10.1200/JCO.2011.38.7589.

R12. Mukaka M, White SA, Terlouw DJ, Mwapasa V, Kalilani-Phiri L, Faragher EB. Is using multiple imputation better than complete case analysis for estimating a prevalence (risk) difference in randomized controlled trials when binary outcome observations are missing? Trials. 2016;17: 341. doi: 10.1186/s13063-016-1473-3.

R13. Power MJ, Freeman C. A Randomized Controlled Trial of IPT Versus CBT in Primary Care: With Some Cautionary Notes About Handling Missing Values in Clinical Trials. Clin Psychol Psychother. 2012;19: 159-169. doi: 10.1002/cpp.1781.

R14. Kunzmann K, Wernisch L, Richardson S, Steyerberg EW, Lingsma H, Ercole A, et al. Imputation of Ordinal Outcomes: A Comparison of Approaches in Traumatic Brain Injury. J Neurotrauma. 2021;38: 455-463. doi: 10.1089/neu.2019.6858.

R15. Farzaneh N, Williamson CA, Gryak J, Najarian K. A hierarchical expert-guided machine learning framework for clinical decision support systems: an application to traumatic brain injury prognostication. NPJ Digit Med. 2021;4: 78. doi: 10.1038/s41746-021-00445-0.

---

## [Decision Letter · Decision Letter 1]

22 Jun 2022

The leap to ordinal: detailed functional prognosis after traumatic brain injury with a flexible modelling approach

PONE-D-22-05175R1

Dear Dr. Bhattacharyay,

We’re pleased to inform you that your manuscript has been judged scientifically suitable for publication and will be formally accepted for publication once it meets all outstanding technical requirements.

Kind regards,

Soojin Park, M.D.

Academic Editor

PLOS ONE

Additional Editor Comments (optional):

Thank you for patience. Despite giving the second reviewer adequate time to respond, they have declined. Based on Reviewer 1 and this editors review of your responses, we recommend Accept.

Reviewers' comments:

Reviewer's Responses to Questions

**Comments to the Author**

1. If the authors have adequately addressed your comments raised in a previous round of review and you feel that this manuscript is now acceptable for publication, you may indicate that here to bypass the “Comments to the Author” section, enter your conflict of interest statement in the “Confidential to Editor” section, and submit your "Accept" recommendation.

Reviewer #1: All comments have been addressed

2. Is the manuscript technically sound, and do the data support the conclusions?

Reviewer #1: Yes

3. Has the statistical analysis been performed appropriately and rigorously? 

Reviewer #1: Yes

4. Have the authors made all data underlying the findings in their manuscript fully available?

Reviewer #1: Yes

5. Is the manuscript presented in an intelligible fashion and written in standard English?

Reviewer #1: Yes

6. Review Comments to the Author

Reviewer #1: (No Response)

7. PLOS authors have the option to publish the peer review history of their article (what does this mean?). If published, this will include your full peer review and any attached files.

Reviewer #1: No

---

## [Editor Report · Acceptance letter]

23 Jun 2022

PONE-D-22-05175R1 

The leap to ordinal: detailed functional prognosis after traumatic brain injury with a flexible modelling approach 

Dear Dr. Bhattacharyay:

I'm pleased to inform you that your manuscript has been deemed suitable for publication in PLOS ONE. Congratulations! Your manuscript is now with our production department. 

Kind regards, 

on behalf of

Dr. Soojin Park 

Academic Editor

PLOS ONE